# Structures of human organellar SPFH protein complexes

Jingjing Gao[1], Dawafuti Sherpa[1], Nikita Kupko [1], Haruka Chino[1], Jianwei Zeng [1] & Sichen Shao [1,2] ✉

Stomatin, Prohibitin, Flotillin, and HflK/C (SPFH) family proteins are found in all kingdoms of life and in multiple eukaryotic organelles. SPFH proteins assemble into homo- or hetero-oligomeric rings that form domed structures. Most SPFH assemblies also abut a cellular membrane, where they are implicated in diverse functions ranging from membrane organization to protein quality control. However, the precise architectures of different SPFH complexes remain unclear. Here, we report single-particle cryo-EM structures of the endoplasmic reticulum (ER)-resident Erlin1/2 complex and the mitochondrial prohibitin (PHB1/2) complex, revealing assemblies of 13 heterodimers of Erlin1 and Erlin2 and 11 heterodimers of PHB1 and PHB2, respectively. We also describe key interactions underlying the architecture of each complex and conformational heterogeneity of the PHB1/2 complex. Our findings elucidate the distinct stoichiometries and properties of human organellar SPFH complexes and highlight common principles of SPFH complex organization.

Proteins in the SPFH family exist in all kingdoms of life and are characterized by a distinctive domain organization that includes at least one evolutionarily conserved SPFH domain named after representative family members: stomatin, prohibitin, flotillin, and HflK/C[1,2]. SPFH proteins assemble into ringed complexes that resemble molecular cages, and most SPFH complexes localize to cellular membranes[2–7]. In addition, all known SPFH complexes are either oligomers of the same SPFH protein, or assemblies of multiple heterodimers of two related SPFH paralogs. SPFH proteins are linked to fundamental cell biological processes, and many are important for human health[1,2,8,9]. However, the precise stoichiometries and molecular functions of different SPFH complexes are incompletely defined.

The best understood SPFH complex comprises the founding prokaryotic SPFH proteins, HflK and HflC. HflK and HflC form heterodimers in the periplasm of gram-negative bacteria, where they are each anchored in the inner membrane via an N-terminal transmembrane helix. Structural studies using single-particle cryogenic microscopy (cryo-EM) have revealed that 12 HflK/C heterodimers (24 subunits in total) assemble to form a cage abutting the membrane[4,10] that may sample different conformations[11]. The HflK/C complex additionally interacts with and modulates the activity of

FtsH, a homohexameric transmembrane complex with cytosolic AAA protease domains[4,10–12]. Thus, it is postulated that HflK/C-FtsH assemblies regulate proteolysis, notably of membrane proteins, in bacteria.

In eukaryotes, SPFH proteins are present both in the cytosol and within membrane-bound organelles. The structures of two SPFH complexes that reside in the eukaryotic cytosol have been determined. The first was the vault, a ~13 MDa assembly of 78 copies of the major vault protein (MVP) resulting from a base-to-base interaction of two cage-like rings, each with 39 MVP subunits[13–16]. Recent cryo-EM reconstructions have also indicated that single cages of 39 MVP may exist as 'half vaults' that exhibit conformational heterogeneity[17]. The vault, whose purpose remains mysterious, appears to be an exception among the SPFH complexes studied to date in its ability to form a complete enclosure and the absence of membrane association. The structure of a second mammalian cytosolic SPFH complex, determined more recently, was that of flotillin[5,6]. This revealed a complex comprising 22 heterodimers of FLOT1 and FLOT2 (44 subunits in total). Both FLOT1 and FLOT2 lack a transmembrane domain but peripherally associate with the cytoplasmic leaflet of the plasma membrane and of endo-secretory vesicles through a hydrophobic interface on their

[1]Department of Cell Biology, Harvard Medical School, Boston, MA, US. [2]Howard Hughes Medical Institute, Boston, MA, US.
✉e-mail: sichen_shao@hms.harvard.edu

N-terminal SPFH domain and post-translational lipid modifications, such as palmitoylation and myristoylation[18].

Although the SPFH complexes visualized to date share similar assembly principles, they differ in overall dimensions and subunit stoichiometries, which were only revealed through structure determination. Hence, the compositions and architectures of other SPFH complexes, including all SPFH complexes known to assemble inside eukaryotic membrane-bound organelles, are unknown. Among these are the organellar SPFH complexes formed by the Erlin proteins (Erlin1 and Erlin2) at the endoplasmic reticulum (ER) membrane and by the prohibitin proteins (PHB1 and PHB2) at the inner mitochondrial membrane. Unlike the vault and flotillin complexes, these SPFH complexes do not reside in the cytosol. The Erlin1/2 complex assembles inside the ER lumen, while the PHB1/2 complex assembles in the mitochondrial intermembrane space. Both complexes are important for human health and cellular homeostasis. Mutations in Erlin1 and Erlin2 are linked to hereditary spastic paraplegias[19–24] and related motor neuron disorders[25–27], while the dysregulation of prohibitin levels is implicated in a wide range of diseases, including cancers, obesity and diabetes, and aging-related neurodegenerative disorders[28–32].

The functions of organellar SPFH complexes remain imprecisely defined. However, compelling hypotheses propose roles for the Erlin1/2 and PHB1/2 complexes in nascent membrane protein quality control at the ER and the inner mitochondrial membrane, respectively. Conceptually, these two SPFH complexes are the most analogous to the prokaryotic HflK/C complex for several reasons. First, PHB1 and PHB2 are the closest eukaryotic homologs to HflK and HflC and have been reported to interact with mitochondrial matrix AAA proteases[33], similar to the HflK/C-FtsH interaction[12]. These PHB1/2 supercomplexes have been implicated in the stability of nascent and unassembled inner mitochondrial membrane proteins[33–35]. Similarly, at the ER, the Erlin1/2 complex has been reported to interact with factors that mediate ER-associated degradation (ERAD), the major ER protein quality control pathway during which substrate proteins are selectively ubiquitylated and then extracted by the p97/Cdc48 AAA motor into the cytosol for proteasomal degradation[36–40]. In association with ERAD factors such as

the ubiquitin ligase RNF170 and the putative p97 adaptor TMUB1, the Erlin1/2 complex has been proposed to facilitate the ERAD of some membrane proteins[38–43]. Like the HflK/C complex, the Erlin1/2 and the PHB1/2 complexes also assemble on the opposite side of the membrane as cytosolic or mitochondrial ribosomes, respectively, engaged in membrane protein synthesis, and are thus well-positioned to engage nascent membrane proteins.

In this study, we report single-particle cryo-EM structures of the human Erlin1/2 and PHB1/2 complexes. Our findings elucidate the distinct stoichiometries and molecular organizations of the primary organellar SPFH complexes, which will accelerate investigations into their mechanism and interactions with putative substrates and

### Table 1 | Cryo-EM Data Collection, Refinement, and Validation

| | Erlin1/2 PDB 9O9U EMD-70263 | PHB1/2 (closed) PDB 9O9Z EMD-70267 | PHB1/2 (open) PDB 9OAO EMD-70268 |
|---|---|---|---|
| **Data collection and processing** | | | |
| Magnification | 105,000 | 165,000 | 165,000 |
| Voltage (kV) | 300 | 300 | 300 |
| Electron exposure (e–/Å²) | 53 | 50 | 50 |
| Defocus range (μm) | -1.2 to -2.2 | -0.8 to -2.0 | −0.8 to −2.0 |
| Pixel size (Å) | 0.83 | 0.73 | 0.73 |
| Symmetry imposed | C1 | C11 | C11 |
| Initial particle images (no.) | 708,455 | 109,090 | 109,090 |
| Final particle images (no.) | 95,469 | 22,299 | 7,010 |
| Map resolution (Å) | 3.0 | 2.4 | 3.1 |
| FSC threshold | 0.143 | 0.143 | 0.143 |
| Map resolution range (Å) | 2.9 to 28.7 | 2.0 to 30.0 | 2.5 to 29.7 |
| **Refinement** | | | |
| Initial model used (PDB code) | ModelAngelo | ModelAngelo | ModelAngelo |
| Model resolution (Å) | 3.3 | 2.6 | 3.5 |
| FSC threshold | 0.5 | 0.5 | 0.5 |
| Model resolution range (Å) | 3.3 to 84.2 | 2.6 to 42.6 | 3.5 to 47.6 |
| Map sharpening *B* factor (Å²) | −61.3 | −44.7 | −41.8 |
| Model composition | | | |
| Non-hydrogen atoms | 60,697 | 43,736 | 43,659 |
| Protein residues | 7,553 | 5,544 | 5,533 |
| Ligands | 26 NAG | | |
| *B* factors (Å²) | | | |
| Protein | 106.55 | 54.50 | 139.76 |
| Ligand | 138.25 | | |
| R.m.s. deviations | | | |
| Bond lengths (Å) | 0.003 | 0.004 | 0.003 |
| Bond angles (°) | 0.474 | 0.482 | 0.455 |
| Validation | | | |
| MolProbity score | 1.71 | 1.25 | 1.98 |
| Clashscore | 6.66 | 4.18 | 8.85 |
| Poor rotamers (%) | 1.73 | 0.96 | 2.52 |
| Ramachandran plot | | | |
| Favored (%) | 97.12 | 97.80 | 96.76 |
| Allowed (%) | 2.88 | 2.2 | 3.24 |
| Disallowed (%) | 0 | 0 | 0 |

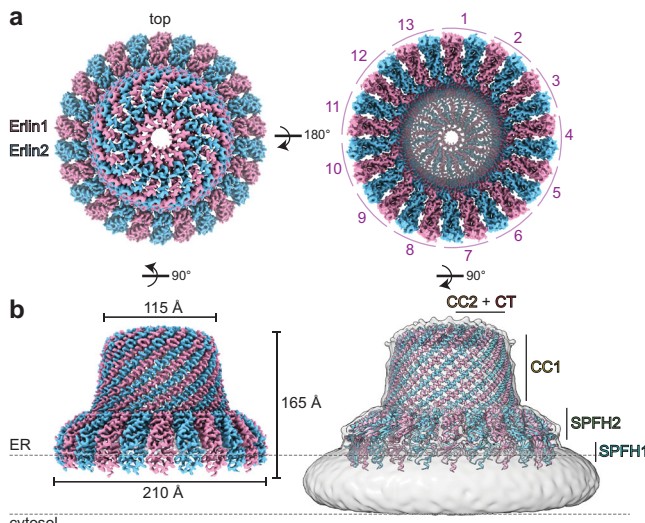

**Fig. 1 | Cryo-EM structure of the Erlin1/2 complex. a** Cryo-EM map of the Erlin1/2 complex, with alternating Erlin1 and Erlin2 subunits colored pink and blue, respectively, viewed from the ER lumen ("top", left) and from the ER membrane (right). **b** Side view of the cryo-EM map (left) and model of the Erlin1/2 complex superposed in a low-pass filtered map showing the detergent micelle (right) as a proxy for the ER membrane (dashed lines). The dimensions of the complex and placement of Erlin1/2 domains are indicated.

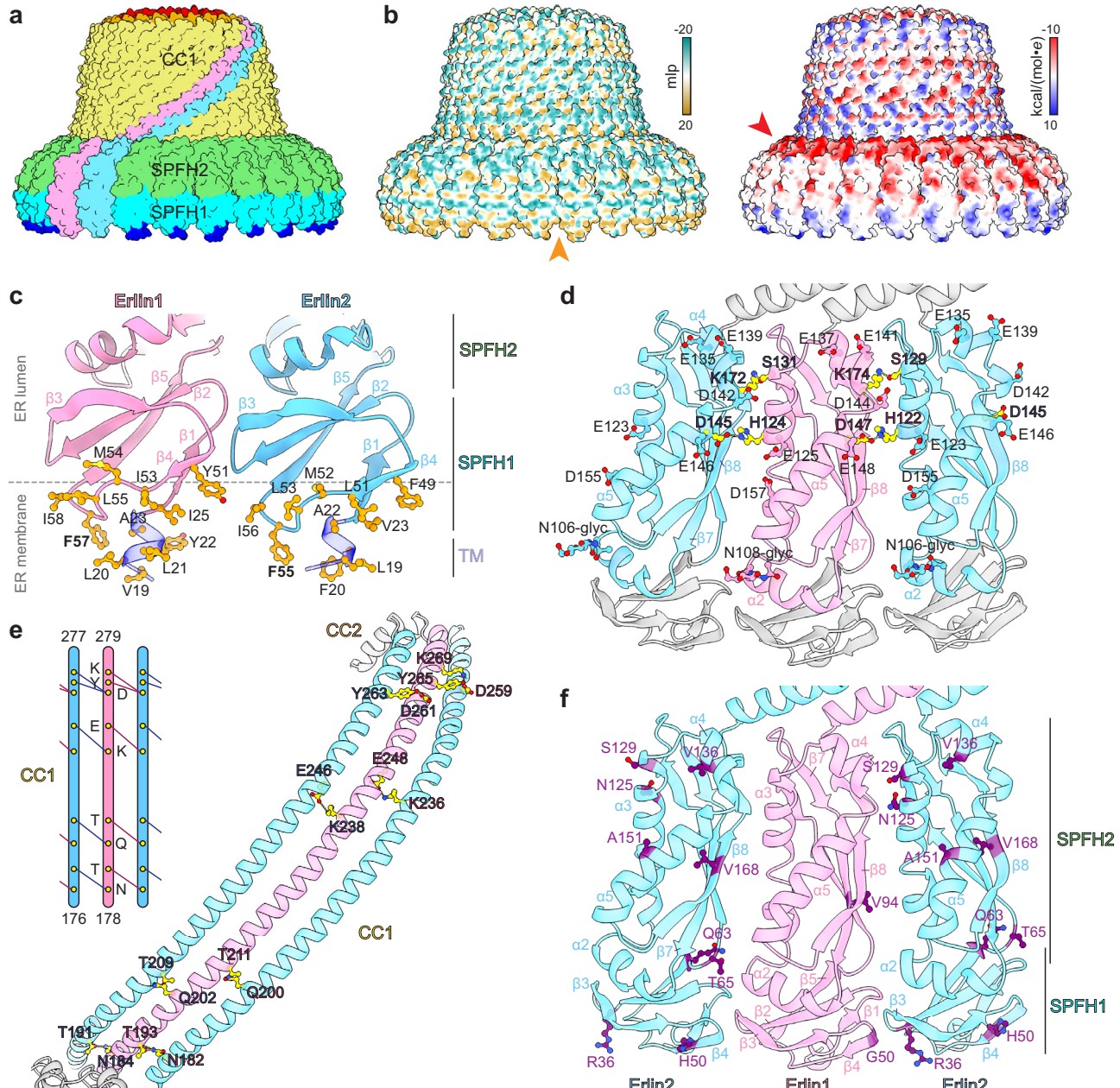

**Fig. 2 | SPFH and CC1 domain interactions in the Erlin1/2 complex. a** Surface representation of the Erlin1/2 complex model, with two individual subunits colored in pink and blue, and the other subunits colored according to the indicated domain. **b** Surface representations of the Erlin1/2 complex model colored by molecular lipophilicity potential (mlp: teal – most hydrophilic, dark goldenrod – most lipophilic, left) or by Coulombic electrostatic potential (right). Orange and red arrowheads note hydrophobic and acidic surfaces. **c** Model of the N-terminal transmembrane (TM) and SPFH1 domains of Erlin1 (pink) and Erlin2 (blue) with hydrophobic residues at the endoplasmic reticulum (ER) membrane interface indicated. **d** Model of alternating Erlin1 and Erlin2 subunits with the SPFH2 domains colored as in (**c**). Residues involved in intersubunit hydrogen bonds are colored yellow and indicated. Residues that contribute to an acidic patch and N-linked glycosylation sites (N-glyc) are also indicated. **e** Model of alternating Erlin1 and Erlin2 CC1 helices with hydrogen bonding interactions indicated. **f** Model as in (**d**) with the positions of disease-linked mutations indicated in purple.

functional partners. Comparisons of the structures of different SPFH complexes also highlight the structural diversity of SPFH complexes despite the common domain organization of this protein family.

## Results

### Cryo-EM structure of the ER-resident Erlin1/2 complex

We first analyzed Erlin1 and Erlin2, which are the only SPFH proteins known to reside in the ER, the primary site for the synthesis of membrane, secreted, and endo-secretory pathway organellar proteins in eukaryotic cells. The two Erlin paralogs are highly similar (74%

identical) and are each predicted to have an N-terminal transmembrane helix followed by tandem SPFH1 and SPFH2 domains, two helices (CC1 and CC2) predicted to form coiled coil interactions, and a C-terminal region (CT) that resides in the ER lumen (Supplementary Fig. 1a, b). Of these regions, the CTs are the most divergent between Erlin1 and Erlin2.

To purify the Erlin1/2 complex, we co-expressed N-terminally Strep-tagged Erlin1 and C-terminally Flag-tagged Erlin2 in Expi293F cells for tandem affinity purification from detergent-solubilized membrane fractions (Supplementary Fig. 1c). Peak fractions after

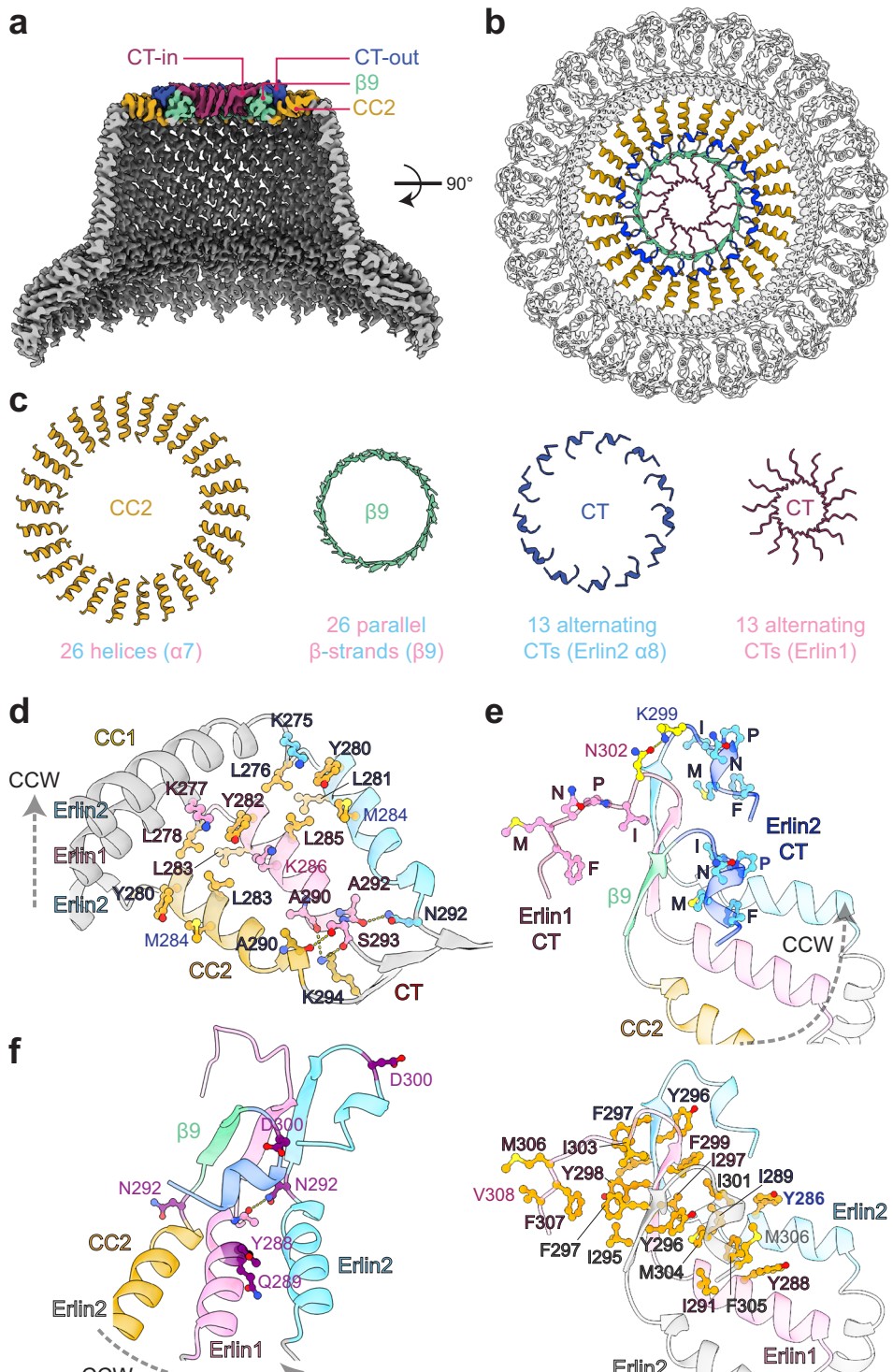

**Fig. 3 | The C-terminal region of the Erlin1/2 complex forms a four-layered structure. a** Clipped side view of the Erlin1/2 complex cryo-EM map with the elements that form four structural layers at the narrow end of the 'cage' colored as indicated. **b** Top view of the Erlin1/2 complex model with the narrow end colored as in (**a**). **c** Isolated elements contributing to the four layers are colored as in (**a**). **d** View of 3 subunits of the Erlin1/2 complex showing hydrophobic packing and hydrogen bonding interactions involving the CC2 helices that facilitate the transitions from the CC1 helices and into β9 of the CTs. Residues outlined are identical between Erlin1 and Erlin2. Arrow indicates counter-clockwise (CCW) direction, when viewed from the top of the complex. **e** The C-terminal organization of 3 subunits in the Erlin1/2 complex, showing the IPNMF motifs, a potential hydrogen bond between the Erlin1 and Erlin2 CTs (top), and hydrophobic packing interactions (bottom). **f** Positions of disease-linked mutations (purple) in the CC2 and CT domains of Erlin1 (pink) or Erlin2 (colored by domain or blue).

size exclusion chromatography were combined and concentrated for cryo-EM analyses (Supplementary Fig. 1d). 2D class averages showing top views of the complex depicted a circular structure with 26 subunits (Supplementary Fig. 2). Because initial 3D reconstructions did not

reveal any distinguishing characteristics between the 26 subunits in the complex, we applied C26 symmetry to generate a consensus refinement and performed a C26 symmetry expansion on the aligned particles (Supplementary Fig. 2). We then performed a focused 3D

classification without alignments using a mask covering the C-terminal end of the complex. This revealed a reconstruction in which we could trace the polypeptide backbone of the CTs of alternating subunits. Refinement of the particles in this class without any applied symmetry resulted in a map with an overall resolution of 3.0 Å (Fig. 1, Table 1, and Supplementary Figs. 2 and 3).

The overall architecture of the Erlin1/2 complex is a domed cage-like structure with a height of ~165 Å extending into the ER lumen (Fig. 1). The wide end formed by the SPFH domains is ~210 Å in diameter, while the narrow end containing the CTs in the ER lumen is ~115 Å in diameter. To date, all structures of heterodimeric SPFH complexes, such as the HflK/C[4,10,11] and FLOT1/2[5,6] complexes, show an alternating arrangement of the two paralogs. In addition, our tandem affinity purification isolated complexes containing both Erlin1 and Erlin2 (Supplementary Fig. 1c). We therefore modeled 13 heterodimeric pairs of Erlin1 and Erlin2 based on the hypothesis that the predominant arrangement of the Erlin1/2 complex would also follow this pattern. We could build a model for the entire complex except for the extreme N- and C-termini of each Erlin protein (residues 1-18 and 310-348 of Erlin1 and residues 1-17 and 308-339 of Erlin2) and could also visualize and build N-linked glycosylation modifications on N108 of Erlin1 and N106 of Erlin2 (Supplementary Fig. 3e).

Although the high sequence and structural similarity between Erlin1 and Erlin2 (Supplementary Fig. 1b) precluded our ability to unambiguously distinguish between the two paralogs in the cryo-EM map, our assignment of alternating Erlin1 and Erlin2 subunits is supported by a concurrent structural study[44] and evidence suggesting that Erlin1 CTs extend toward the center of the cage while alternating Erlin2 CTs turn back toward the outer edge of the cage (discussed below). However, we cannot rule out the possibility that Erlin1/2 complexes may include a mixture of subunit arrangements, which may be consistent with observations that the ratio of the two paralogs can vary in complexes and that Erlin1 and Erlin2 can each form higher-order complexes, albeit with distinct biochemical properties, in the absence of the other paralog[45,46] (Supplementary Fig. 4a, b). Hence, most of the intersubunit interactions described below could occur at heterotypic Erlin1-Erlin2 or Erlin2-Erlin1 interfaces, as well as monotypic Erlin1-Erlin1 or Erlin2-Erlin2 interfaces.

## Key interactions within the Erlin1/2 complex

In addition to the N-terminal transmembrane domain, hydrophobic and aromatic residues in the SPFH1 domain of Erlin1 and Erlin2 likely contribute to complex association with the lumenal face of the ER membrane (Fig. 2a-c). We also observed additional density associated with the SPFH1 domain of each subunit that appears to be coordinated by several polar residues (Erlin1 Y36 and R38 or Erlin2 Y34 and R36) and the backbone of the β4-β5 loop (Supplementary Fig. 3f). This density may be consistent with the binding of phospholipid headgroups[44]. The interface between adjacent SPFH2 domains is stabilized by two hydrogen bonds: one between a histidine (Erlin1 H124 or Erlin2 H122) and an aspartate (Erlin1 D147 or Erlin2 D145), and another between a serine (Erlin1 S131 or Erlin2 S129) and a lysine (Erlin1 K174 or Erlin2 K172) (Fig. 2d). Following the SPFH2 domain, the 101-residue CC1 helices interlock through hydrogen bonding and other electrostatic interactions to produce a sealed wall of 26 helices (Fig. 2e). Interestingly, while many disease-linked mutations in Erlin1 and Erlin2 map to the SPFH domains (Fig. 2f; discussed below), no reported disease-linked mutations map to the CC1 region.

The narrow end of the Erlin1/2 cage forms a 4-layer structure (Fig. 3a–c). The CC2 layer contains 26 CC2 helices that each bend ~80° from the CC1 helix towards the center of the cage. This arrangement is stabilized through hydrophobic stacking interactions mediated by a leucine (Erlin1 L285 or Erlin2 L283) in the CC2 helix of one subunit and two leucines (Erlin1 L278 and L283 and Erlin2 L276 and L281) on the CC1-CC2 linker and CC2 helix of the subunit in the counter-clockwise

direction, when viewed from the top of the complex (Fig. 3d). A tyrosine (Erlin1 Y282 or Erlin2 Y280) may also facilitate this transition through cation-π interactions with a lysine (Erlin1 K277 or Erlin2 K275) on the CC1 helix.

Next, a hydrogen bonding network facilitates the transition from the layer of 26 CC2 helices into a layer of 26 parallel β-strands contributed by β9 of each subunit (Fig. 3d). This network involves a conserved stretch of residues (AIASNSK: residues 290-296 of Erlin1 or residues 288-294 of Erlin2) that span the linker between the CC2 helix and β9 (Fig. 3d and Supplementary Fig. 1b). The sequences of the CC2-β9 linker and of β9 are identical between Erlin1 and Erlin2 (Supplementary Fig. 1b). However, the hydrogen bonding interactions result in a 1-residue offset between adjacent β9 strands, such that the side-chains of the corresponding residues on neighboring subunits reside on opposite sides of the ring formed by the parallel β9 strands (Fig. 3e and Supplementary Fig. 4c).

After β9, the CTs of alternating Erlin subunits extend in opposite directions (Fig. 3a, b and Supplementary Fig. 4d). One CT (modeled as Erlin2) forms a short helix and extends towards the outer edge of the cage. This CT conformation is seen on subunits in which the sidechain of the tyrosine in β9 (Erlin1 Y298 or Erlin2 Y296) resides at the outer surface of the β9 ring (Fig. 3e and Supplementary Fig. 4c, d). The other CT (modeled as Erlin1) extends a flexible segment towards the inside of the cage, leading to a central channel-like structure with an inner diameter of ~12 Å (Supplementary Fig. 4e). This transition is facilitated by two residues after β9 that are divergent between Erlin1 and Erlin2 (Erlin1 S301 and N302, and Erlin2 K299 and D300) (Supplementary Fig. 1b). Notably, a hydrogen bond can form between Erlin1 N302 and Erlin2 K299 (Fig. 3e) that would only be possible at interfaces between Erlin1 and Erlin2 or between two Erlin2 subunits, in which case D300 of one Erlin2 subunit could hydrogen bond with K299 of the other. Because this putative interaction resides at the turning point of the CTs in opposite directions, we hypothesize that these interactions may favor the Erlin2 CT to turning back towards the outer edge of the complex.

The Erlin CTs, in particular a conserved hydrophobic stretch (IPNMF: residues 303-307 of Erlin1 or residues 301-305 of Erlin2) immediately following the two divergent residues discussed above, have been reported to be essential for the higher-order assembly of the Erlin1/2 complex[45]. Our structure revealed that these residues in both CT conformations are engaged in extensive hydrophobic packing interactions (Fig. 3e). Notably, the isoleucine, methionine, and phenylalanine in the IPNMF motif of the CT that extends outwards (modeled as Erlin2) interact with CC2 and β9 of the heterodimer in the counter-clockwise direction, when viewed from the top of the cage. Specifically, these residues pack with a tyrosine (Erlin1 Y298 or Erlin2 Y296) on β9 of the same subunit, the outward-pointing phenylalanine (Erlin1 F299 or Erlin2 F297) on β9 of the adjacent subunit, and a tyrosine and isoleucine (Erlin1 Y288 and I291 or Erlin2 Y286 and I289) on the CC2 helices of both subunits in the neighboring heterodimer.

Similarly, the isoleucine, proline, and phenylalanine residues in the IPNMF motif of the CT that extends inwards (modeled as Erlin1) form hydrophobic interactions with β9 of the same subunit as well as with the β9 strands of both subunits in the heterodimer in the counterclockwise direction (Fig. 3e). In addition, at the interior of the cage, the methionine of the IPNMF motif and the residue following the motif (Erlin1 V308 or Erlin2 M306) interlock with adjacent hydrophobic residues, which may form a hydrophobic surface along the central 'cavity' of the cage (Supplementary Fig. 4e, f).

## Positions and impacts of Erlin mutations

Mutations in Erlin1 and Erlin2 are linked to the development of hereditary spastic paraplegias SPG18 and SPG62 and amyotrophic lateral sclerosis (ALS)[19-27]. Mapping the positions of disease-linked mutations to our structure revealed that they fall into three locations (Figs. 2f and

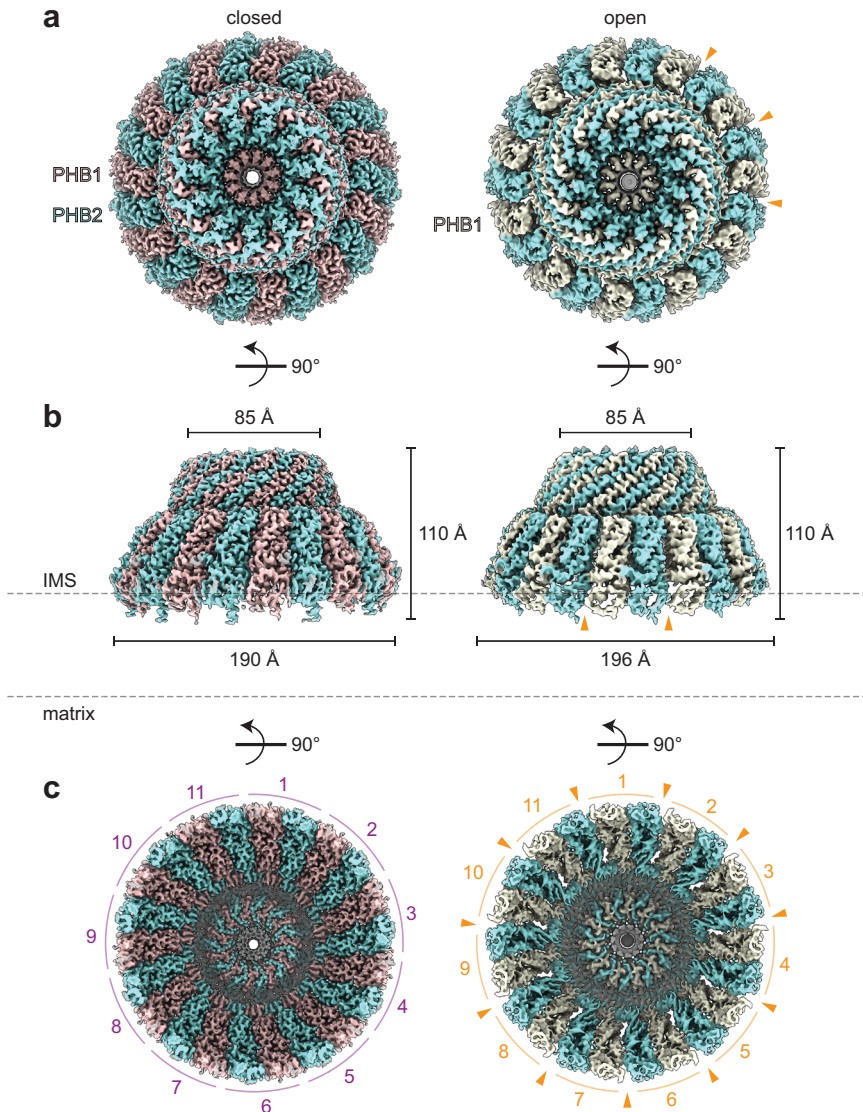

**Fig. 4 | Cryo-EM structures of the PHB1/2 complex. a** "Top" views of cryo-EM maps of the closed (left) and open (right) conformations of 11 PHB1 (mauve, left or cream, right) and PHB2 (robin blue) heterodimers. **b** Side views of the maps as in (**a**), with the dimensions of each complex indicated. IMS, intermembrane space **c** Cryo-EM maps as in (**a**), viewed from the inner mitochondrial membrane, with the distinct PHB1/2 heterodimers indicated. Orange arrowheads indicate gaps visible in the open conformation.

3f). First, G50 of Erlin1 and R36 and H50 of Erlin2 reside at the ER membrane interface[23,47] (Fig. 2c, f). Notably, G50 of Erlin1 interacts with the β2-β3 loop of the adjacent subunit (Fig. 2d, f). Secondly, V94 of Erlin1, and Q63, T65, N125, S129, V136, A151, and V168 of Erlin2 reside in the SPFH2 domains[20,25,26,46–48] (Fig. 2f). Among these, T65, N125, and S129 of Erlin2 contribute to stabilizing the intersubunit interface, while the other residues contribute to the packing of the hydrophobic core of the SPFH2 domain (Fig. 2d). Mutations in the SPFH domains may also inhibit interactions with proposed binding partners of the Erlin1/2 complex[46]. Thirdly, Y288 and Q289 of Erlin1 and N292 and D300 of Erlin2 reside at the narrow end of the cage[23,26,47] (Fig. 3f). Y288 participates in hydrophobic packing with outward-extending CTs of neighboring subunits (Fig. 3e). N292 of Erlin2 resides in the conserved linker between CC2 and β9, where its sidechain may hydrogen bond with the backbone of an alanine (Erlin1 A292 or Erlin2 A290) in the CC2 helix of the subunit in the clockwise direction, when viewed from the top of the cage (Fig. 3d). Finally, D300 of Erlin2 is one of the two residues following β9 that are different in Erlin1 and that may facilitate CT placement (Fig. 3e, f and Supplementary Fig. 1b). Hence,

many disease-linked mutations may interfere with Erlin1/2 complex assembly.

To test how interactions observed in our structure contribute to complex assembly, we performed mutagenesis (Supplementary Fig. 5). Blue Native polyacrylamide gel electrophoresis (BN-PAGE) revealed that Erlin1 and Erlin2 migrate in the ~900 kDa range, consistent with a 26-subunit complex (Supplementary Fig. 4a). In Erlin1 knockout (KO) cells, Erlin2 could still form ~900 kDa assemblies but was additionally detected in lower molecular weight ranges, which may reflect impaired assembly intermediates (Supplementary Fig. 4a, b). Conversely, in Erlin2 KO cells, Erlin1 migrated at a higher molecular weight, over ~1 MDa (Supplementary Fig. 4a, b). We therefore tested if re-expressing Erlin1 or Erlin2 variants in the respective KO cells could rescue these signatures of altered complex assembly (Supplementary Fig. 5a–c). Importantly, re-expressing wildtype (WT) Erlin1 in Erlin1 KO cells decreased the abundance of lower molecular weight Erlin2 complexes (Supplementary Fig. 5b), and similarly, re-expressing WT Erlin2 in Erlin2 KO cells 'rescued' the migration of Erlin1 back to the ~900 kDa region (Supplementary Fig. 5c).

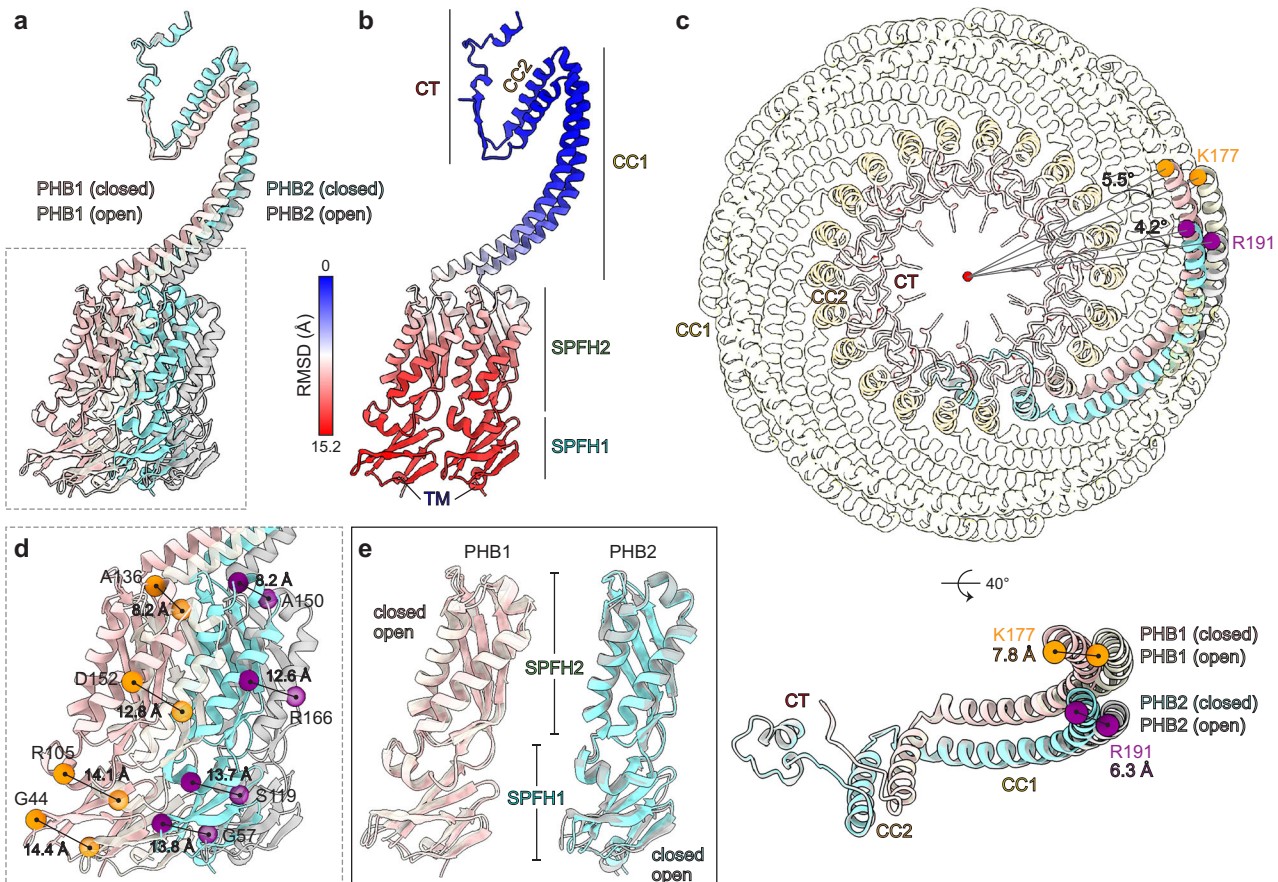

**Fig. 5 | Comparisons of the closed and open PHB1/2 complexes. a** Superposition of a PHB1/2 heterodimer in the closed (mauve and robin blue) and open (cream and gray) PHB1/2 complex conformations. **b** Model of the PHB1/2 heterodimer in the closed complex conformation as in (**a**), colored by root mean square deviation (RMSD) values compared to the open complex. Domain features are indicated. **c** 'Bottom' view (top) of the closed and open PHB1/2 complex models with one heterodimer in each complex, colored as in (**a**) and the others colored according to domain, and a rotated view showing only an isolated heterodimer (bottom). The rotation, relative to the centroid of the complex (top), and the translation (bottom) of the N-terminal residue of the PHB1 (K177, orange) or PHB2 (R191, purple) CC1 helix from the closed (mauve and robin blue) to open (cream and gray) conformation are indicated. **d** Zoomed inset of the superposition as in (**a**), showing the distance shifted by the indicated PHB1 (orange) or PHB2 (purple) residues in the SPFH domains between the closed and open conformations. **e** Superposition of the isolated SPFH1 and SPFH2 domains of PHB1 and PHB2 in the closed and open PHB1/2 complex conformations, colored as in (**a**).

For both Erlin1 and Erlin2, we tested analogous mutants (Supplementary Fig. 5a). In the 'M1' variants, the histidine and serine residues that mediate hydrogen bonding between SPFH2 domains were mutated to alanines (Fig. 2d). The 'M2' variants contain four mutations to disrupt hydrophobic packing interactions between the CC2 helices (Fig. 3d). The remaining variants contain mutations in the CTs, either of the IPNMF motif to alanines (M3), or of the paralog-specific residues (Erlin1 S301 and N302 or Erlin2 K299 and D300) to alanines (M4) or to swap amino acids (M5: Erlin1 S301K / N302D or Erlin2 K299S / D300N) (Fig. 3e). For both Erlin1 and Erlin2, the M2 and M3 variants did not assemble into higher-order complexes and failed to rescue the KO signature of the other paralog, highlighting the importance of the CC2 interactions and IPNMF motifs[45] in complex assembly (Supplementary Fig. 5b, c). In contrast, the M1 and M5 variants were nearly indistinguishable from WT, while the M4 variants generally rescued the KO signature of the other paralog but displayed some defects in assembling into ~900 kDa complexes. The severities of these mutant phenotypes were largely mirrored in the ability of tagged versions of each variant to pull down the other paralog (Supplementary Fig. 5d, e). The results obtained with the M4 and M5 variants also suggest that although the polar nature of the paralog-specific CT residues facilitates complex assembly, the exact amino acid identities are not essential for complex assembly, consistent with the observation that these amino acids are not conserved in the two Erlin paralogs across different model organisms (Supplementary Fig. 5f, g).

## Cryo-EM structures of the mitochondrial PHB1/2 complex

We next sought to compare the Erlin1/2 complex with the PHB1/2 complex, the other major organellar SPFH protein complex linked to protein biogenesis and quality control. The PHB1/2 complex resides in the intermembrane space (IMS) of mitochondria, where PHB1 and PHB2, which are 50% identical at the sequence level, are each anchored in the inner mitochondrial membrane via an N-terminal transmembrane helix (Supplementary Fig. 6a, b). This is followed by structurally conserved SPFH1 and SPFH2 domains, two coiled-coil domains CC1 and CC2, and distinct CTs. Notably, the PHB2 CT possesses a short extension (residues 285-299) not present in PHB1 that is predicted to form a helix (Supplementary Fig. 6a, b).

To purify the PHB1/2 complex, we generated an Expi293F cell line stably expressing N-terminally Strep-tagged PHB1 using lentiviral transduction. We then transiently expressed untagged PHB2 in these cells and performed a one-step affinity purification for cryo-EM analysis (Supplementary Fig. 6c, d). This revealed that the PHB1/2 complex forms 22-subunit domed cage-like structures with 11 heterodimers of PHB1 and PHB2 (Fig. 4 and Supplementary Figs. 7–9), matching the stoichiometry concurrently reported of a fungal PHB1/2 complex[49].

Previous studies indicate that the PHB1/2 complex associates with mitochondrial matrix AAA proteases[33]. However, although we can detect mitochondrial AAA protease subunits (AFG3L2 and SPG7) at substoichiometric levels in PHB1/2 purifications by immunoblotting (Supplementary Fig. 6d), the mitochondrial AAA proteases did not stably copurify with PHB1/2 (Supplementary Fig. 6c), even when overexpressed. We also did not observe density consistent with AAA protease complexes in our cryo-EM analysis (Supplementary Fig. 7). AAA protease complexes were similarly not obvious in recent cryo-electron tomography (cryo-ET) studies that visualized prohibitin complexes in the mitochondria of different cell types[49–52], although they could be seen at low resolution in negative stain micrographs of tandem-purified complexes[49]. Thus, AAA protease interactions may be transient, heterogeneous, or occur only with a small subset of PHB1/2 complexes.

## Conformational heterogeneity of the PHB1/2 complex

3D classifications of our cryo-EM dataset revealed two main conformations of the PHB1/2 complex: a 'closed' conformation determined to an overall resolution of 2.4 Å, and an 'open' conformation, determined to an overall resolution of 3.1 Å, that showed increased gaps between the SPFH domains of PHB1/2 heterodimers (Fig. 4, Table 1, and Supplementary Figs. 7–9). We also observed a small percentage of flat 'flower-like' particles with 11 splayed-out 'petals' in the raw micrograph images and 2D class averages. However, these particles resided exclusively in thin ice and are likely an artifact of sample freezing (Supplementary Fig. 7a, b).

The closed and open PHB1/2 complexes both have a height of ~110 Å and a diameter of ~85 Å at the narrow end of the cage. The diameter of the wide end of the closed PHB1/2 complex is ~190 Å, and that of the open PHB1/2 complex is ~196 Å (Fig. 4). These dimensions are consistent with recent subtomogram averages of native prohibitin complexes[49–52]. Two cryo-ET studies also observed closed and more open conformations of the prohibitin complex that revealed 11 'spokes'[51,52]. However, the low-resolution cryo-ET reconstructions precluded unambiguous placement of PHB1 and PHB2 and resulted in differing interpretations regarding the number of subunits in the complex. In comparison, our ability to distinguish between and accurately model PHB1 and PHB2 clearly revealed an arrangement of 11 heterodimers of PHB1 and PHB2, with the CT of PHB1 extending inwards towards the center of the cage and the CT of PHB2 extending outward (Supplementary Figs. 9 and 10a).

The major structural changes between the closed and open PHB1/2 complex conformations occur at the SPFH domains and the adjacent CC1 region (up to PHB1 V200 and PHB2 V214 in the CC1 helix; rmsd = 9.783 Å), while the C-terminal portion of the PHB1/2 complex remains largely unchanged (rmsd = 0.708 Å; Fig. 5a–d). In addition to the increased diameter of the ring formed by the SPFH domains in the open PHB1/2 complex, the diameter of the ring formed by the N-terminal ends of the CC1 helices increases from ~102 Å in the closed conformation to ~114 Å in the open conformation. This is achieved through a ~7.8 Å shift of the N-terminal end of the PHB1 CC1 helix, which corresponds to a clockwise rotation of ~5.5° around the C11 axis when viewed from the 'bottom' of the cage (Fig. 5c). Interestingly, the N-terminal end of the PHB2 CC1 helix undergoes a smaller ~6.3 Å shift, corresponding to a clockwise rotation of ~4.2°. This translates to larger differences in the relative positions of the SPFH domains (Fig. 5a, d), without inducing significant structural changes within the domains themselves (Fig. 5e). The discrepancy in the relative movement of the PHB1 and PHB2 CC1 helices also results in distinct changes in the buried area between each intersubunit interface in the closed and open conformations (Fig. 6a).

## Key interactions within PHB1/2 complexes

Like in the Erlin1/2 complex, hydrophobic and aromatic residues in the SPFH1 domain of both PHB1 and PHB2 contribute to membrane anchoring together with the N-terminal transmembrane helix (Fig. 6b). This hydrophobic interface does not change between the closed and open conformations. In both conformations, we could also observe additional densities near the membrane interface of the SPFH1 domains, including one that may be coordinated by polar residues (N53, Q59, and Q76) on PHB2 and another between PHB1 and PHB2 (Supplementary Fig. 9d). In addition, along the β5-β6 linker of PHB1, which marks the transition between the SPFH1 and SPFH2 domains, we observed additional densities at the intersubunit interface that may involve modification of C69 of PHB1 and that appear to stack with R123 of PHB2 and R35 and R97 of PHB1 (Supplementary Fig. 9d). These densities may reflect specific phospholipid interactions, although the local resolution of our maps, additional classifications, and mass spectrometry analysis for covalent modifications did not reveal specific candidates for the identities of these putative interactors.

In the closed PHB1/2 complex, adjacent SPFH domains interact primarily through hydrogen bonds (Fig. 6c). Most of these interactions occur between β8 of one SPFH2 domain and α3 of the neighboring subunit. The sidechain of a conserved arginine (PHB1 R157 or PHB2 R171) in α5, and backbone hydrogen bonds with residues in α2 and the β5-β6 linker also participate in this interaction network. Hydrophobic stacking between a phenylalanine in α5 (PHB1 F161 or PHB2 F175) and a leucine in β7 (PHB1 L95 or PHB2 Leu109) further stabilizes the interface between neighboring SPFH2 domains. Adjacent SPFH1 domains also interact through hydrogen bonds between the β3-β4 and β2-β3 loops. Most of these SPFH domain interactions are lost in the open PHB1/2 complex (Fig. 6d).

The closed PHB1/2 complex also exhibits an extensive hydrogen bonding network between the β6-β7 loop in the SPFH2 domain of each subunit with the CC1 helix of the subunit in the clockwise direction, when viewed from the top of the cage (Fig. 6c and Supplementary Fig. 10b). Hydrogen bonding between adjacent subunits continue through the length of the 22 CC1 helices (Fig. 6e). Most of the interactions that involve residues at the membrane-proximal end of the CC1 helices, including the hydrogen bonding network with the β6-β7 loops, are disrupted in the open PHB1/2 complex (Fig. 6d, e), consistent with the larger conformational differences at the N-terminal ends of PHB1 and PHB2 between the closed and open conformations (Fig. 5c).

The narrow end of the PHB1/2 cage forms a four-layered structure (Fig. 6f and Supplementary Fig. 10c), like the Erlin1/2 complex. First, the CC1 helix of each PHB1 and PHB2 transitions into the CC2 helix through a sharp ~55° bend that is stabilized through two hydrogen bonds between the CC1 ring and the CC2 ring (Fig. 6g). The first is between R239 in the CC2 helix of PHB1 with E231 in the CC1 helix of the PHB2 subunit in the counter-clockwise direction, when viewed from the top of the cage. The second is between Y248 in the CC2 helix of PHB2 with D217 of the CC1 helix of the PHB1 subunit to the right. The layer of 22 PHB1 and PHB2 CC2 helices is further stabilized through hydrogen bonds. E245 of PHB1 can hydrogen bond with N259 and R252 of the PHB2 subunit to the right, and D233 and D246 of PHB1 can hydrogen bond with K250 and Q258, respectively, of the PHB2 subunit on the other side.

The layer of 22 CC2 helices then transitions into a layer of 22 parallel β-strands formed by β9 of alternating PHB1 and PHB2 subunits. An inter-subunit hydrogen bonding network involving residues in the loop between CC2 and β9 facilitates this transition (Fig. 6g). After the β9 layer, the CTs following β9 of the 11 PHB2 subunits each turn back towards the outer edge of the cage, while the CTs following β9 of the 11 PHB1 subunits extend inwards towards an ~18 Å-wide central channel (Fig. 6f and Supplementary Fig. 10a, d). This architecture also exhibits extensive hydrophobic packing (Fig. 6h), especially between the CC2 helix of one PHB2 and the CT of the PHB2 to the left, which may facilitate the ability of the PHB2 CT to turn after β9 and extend back towards the outer edge of the cage.

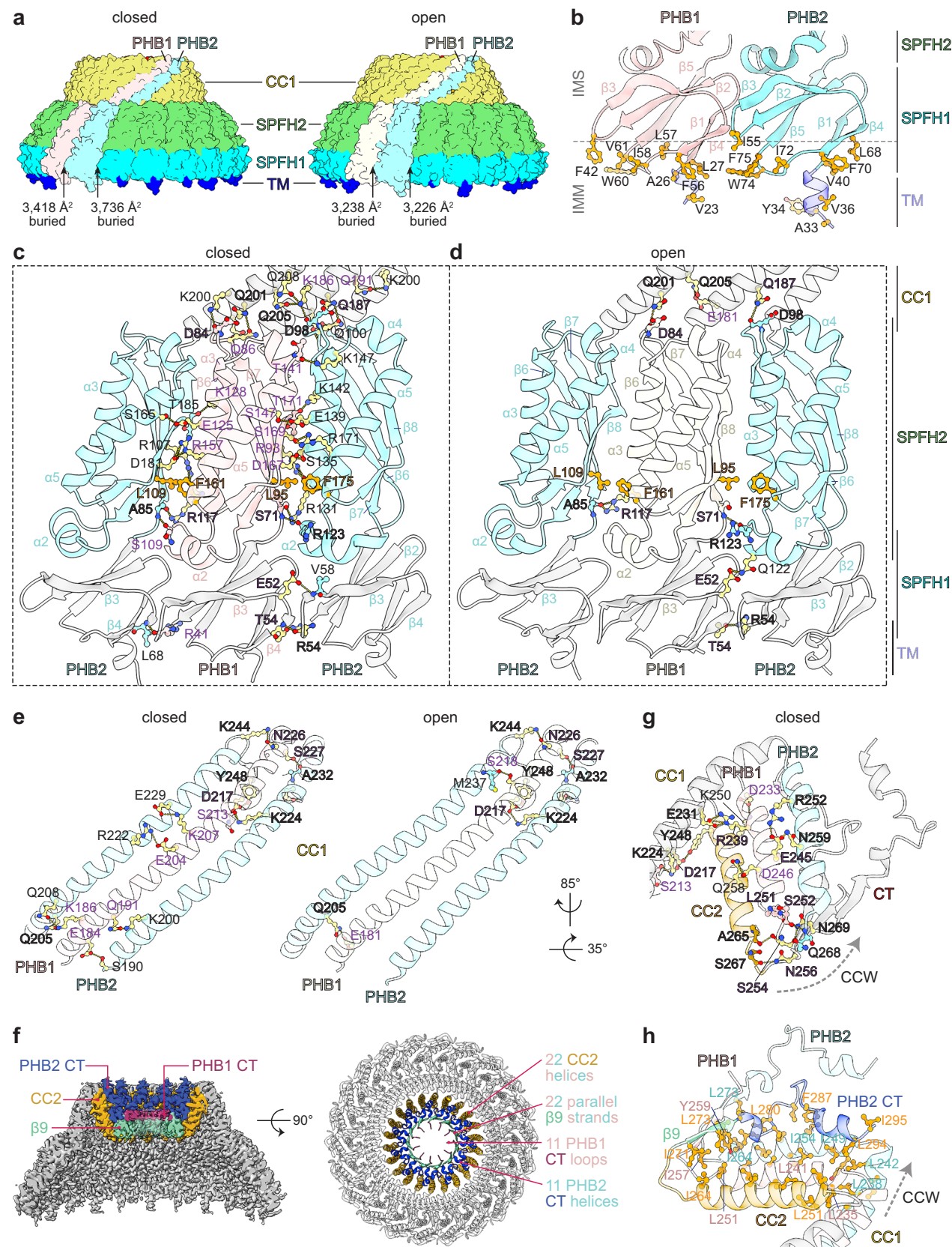

Similar to our mutagenesis analyses of the Erlin1/2 complex (Supplementary Fig. 5), we found that mutations that disrupt interactions in the CC2 helices had the most severe impacts on PHB1/2 complex assembly (Supplementary Fig. 10e, f). Co-expressed N-terminally Strep-tagged PHB1 and C-terminally Flag-tagged PHB2

comigrated in the ~600 kDa range on BN-PAGE, consistent with a 22-subunit complex. Replacing wildtype Strep-PHB1 (Supplementary Fig. 10e) or PHB2-Flag (Supplementary Fig. 10f) with variants carrying mutations in the CC1 helix or the CT to disrupt hydrogen bonding interactions slightly shifted the BN-PAGE migration of the complex.

**Fig. 6 | Interactions in the PHB1/2 complex. a** Surface representation of the closed (left) or open (right) PHB1/2 complex model, with two individual subunits colored as indicated, and the other subunits colored by domain. The buried area at the intersubunit interface on each side of the indicated PHB2 subunit is listed. **b** Model of the N-terminal transmembrane (TM) and SPFH1 domains of PHB1 (mauve) and PHB2 (robin blue) with hydrophobic residues at the inner mitochondrial membrane (IMM) interface indicated. IMS, intermembrane space. **c** Model of alternating PHB1 and PHB2 subunits in the closed PHB1/2 complex with the SPFH2 domains colored as in (**a**). Residues involved in intersubunit hydrogen bonds via their sidechain (yellow) or backbone (mauve or robin blue) are indicated. Residues involved in hydrophobic packing are orange. **d** As in (**c**), but for the open PHB1/2 complex and PHB1 colored cream. Residues outlined maintain interactions in both the closed and open PHB1/2 complex. **e** Model of alternating PHB1 and PHB2 CC1

helices in the closed (left) or open (right) complex, with hydrogen bonding interactions indicated as in (**c** and **d**). **f** Clipped side view of the cryo-EM map (left) or top view of the model (right) of the closed PHB1/2 complex with the elements that form four structural layers at the narrow end of the cage colored as indicated. **g** Hydrogen bonding interactions involving the CC2 domains of PHB1 (mauve) and PHB2 (orange or robin blue), shown on the model of the closed PHB1/2 complex. Arrow indicates counter-clockwise (CCW) direction, when viewed from the top of the complex. **h** Hydrophobic packing interactions involving the CC2 and CT domains of PHB1 (mauve) and PHB2 (colored according to the indicated 4-layer feature or robin blue). Select residues of PHB1 (mauve), the PHB2 subunit in the clockwise direction (orange), or the PHB2 subunit in the counter-clockwise direction (robin blue) are indicated.

However, these variants still efficiently assembled into a ~600-700 kDa complex with the other PHB paralog, based on their comigration. In contrast, introducing mutations into PHB1 predicted to disrupt intersubunit hydrogen bonding interactions at the CC1-CC2 transition destabilized PHB1 (Supplementary Fig. 10e). PHB2 variants carrying mutations in this region were more stable and still assembled into ~600 kDa complexes, but with much less comigration of wildtype PHB1 (Supplementary Fig. 10f). These results suggest that CC2 interactions play an important role in PHB1/2 complex assembly, perhaps particularly for the incorporation of PHB1.

## Comparisons of SPFH complex organizations

Our structures reveal a detailed understanding of the stoichiometry and assembly of two human organellar SPFH protein complexes and add to the handful of experimentally determined SPFH complex structures[4–6,10,13,17,53] (Fig. 7). Comparing these SPFH complexes highlights both common organizational principles and remarkable diversity in their architectures. Although all SPFH complexes share a general domed structure, they vary in subunit copy numbers. Our finding that the PHB1/2 complex contains 22 subunits formed by 11 PHB1/2 heterodimers and that the Erlin1/2 complex contains 26 subunits, putatively formed by 13 Erlin1/2 heterodimers, adds to this diversity. Unsurprisingly, considering the structural conservation of the SPFH domains that form the base of most of these complexes, the number of subunits correlates with the overall dimensions of the complexes, although additional protein domains, such as those found in MVP, or accessory proteins, will also impact complex size (Fig. 7a–c). Notably, the human Erlin1/2 complex has the same number of subunits as the complex formed by 26 copies of the bacterial SPFH protein QmcA[53], and these complexes have similar diameters at the membrane interface (Fig. 7a, b). Moreover, with 22 subunits, the PHB1/2 complex is the most compact SPFH complex observed so far (Fig. 7a, b). Comparing SPFH complexes also revealed that the total number of subunits in SPFH complexes correlates especially well with the length of the CC1 helix (Fig. 7d). This correlation can be used to estimate copy numbers of SPFH proteins that are within two of the actual copy number for all complexes with known structures. It will be interesting to see if this correlation holds as more structures of SPFH complexes are determined.

In addition, while the C-terminal domains of SPFH proteins that form the 'cap' of SPFH complex cages are the most variable regions between the complexes (Fig. 7b), two common principles of their organization are emerging. First, the alternating CTs of heterodimeric SPFH complexes (HflK/C, FLOT1/2, PHB1/2, Erlin1/2) consistently exhibit different conformations, with the CT of one paralog turning back towards the outer edge of the cage and the CT of the other paralog reaching towards the center of the cage (Figs. 3, 6f, and 7). These distinct alternating CT arrangements have also recently been observed in SPFH complexes comprised of a single protein, such as the stomatin and the vault complexes[54,55]. Second, the C-terminal caps of SPFH complexes do not appear to be completely sealed at the center

(Fig. 7b, c and Supplementary Figs. 4e and 10d). Because of limitations owing to lower local resolutions in this region in cryo-EM maps and potential artefacts that may be introduced at the center of the complexes from applying symmetry during data processing, it is not yet clear if these observations represent bona fide 'channels' or if these regions are filled by unstructured C-terminal sequences of SPFH proteins that cannot be modeled into the maps. Interestingly, the outer surface of the 'cap' of flotillin complexes has also been observed to associate with membranes upon the preparation of native membrane vesicles[5], and we can observe positive curvature detergent micelle density associated with the inner surface of the 'cap' of PHB1/2 complexes (Supplementary Fig. 7b and 10d). Whether these observations represent functionally relevant interactions remains to be determined.

## Discussion

Altogether, our findings establish a structural framework for studying the functions of the Erlin and prohibitin complexes. Notably, because AI-based structural predictions fail to confidently assign the stoichiometry of complexes with high copy numbers of repeating subunits, these experimentally determined structures represent the best-validated molecular scaffolds for investigating how functional interactors, such as ERAD factors or mitochondrial proteases, may assemble with Erlin or prohibitin complexes, respectively.

Considering the putative roles of these organellar SPFH complexes in protein quality control, it is also tempting to speculate that the domed cages formed by these complexes become occupied for functional purposes. For example, the periplasmic domains of FtsH engage the SPFH2 domains of HflK/C inside the cage[4,10]. Achieving this interaction may require conformational changes that allow for lateral opening of the SPFH complex[4,11] or the co-assembly of FtsH as HflK/C complexes are formed. The observations that the PHB1/2 complex may exist in both closed and more open conformations in our study (Fig. 4) and by cryo-ET[51,52] implicate another route for membrane proteins to access the interior of SPFH complexes. The interiors of SPFH complexes are also reminiscent of chaperonin chambers that facilitate protein folding through sequestration[56], although SPFH complexes lack ATP hydrolysis-regulated access and conformational changes that alter the inner surface properties of chaperonins. Such 'cargo' sequestration has been observed in situ for ribosomes enclosed within vaults[16]. The Erlin1/2 complex has also been suggested to 'anchor' certain transmembrane proteins, such as SREBP and DNAJB12, in the ER[57,58], which prevents these proteins from trafficking to other organelles and impacts the release of viral particles[57–59]. Future analyses of SPFH complex interactions with putative sequestration substrates will advance our understanding of the potential role of the caged architecture in these functions.

Finally, based on their large architectures and membrane association, many SPFH complexes, including Erlin1/2 and PHB1/2, are assumed to play a role in membrane organization, particularly in the formation of microdomains that may have specialized functions[2,3]. Both Erlin1/2 and PHB1/2 have been reported to bind specific lipids and

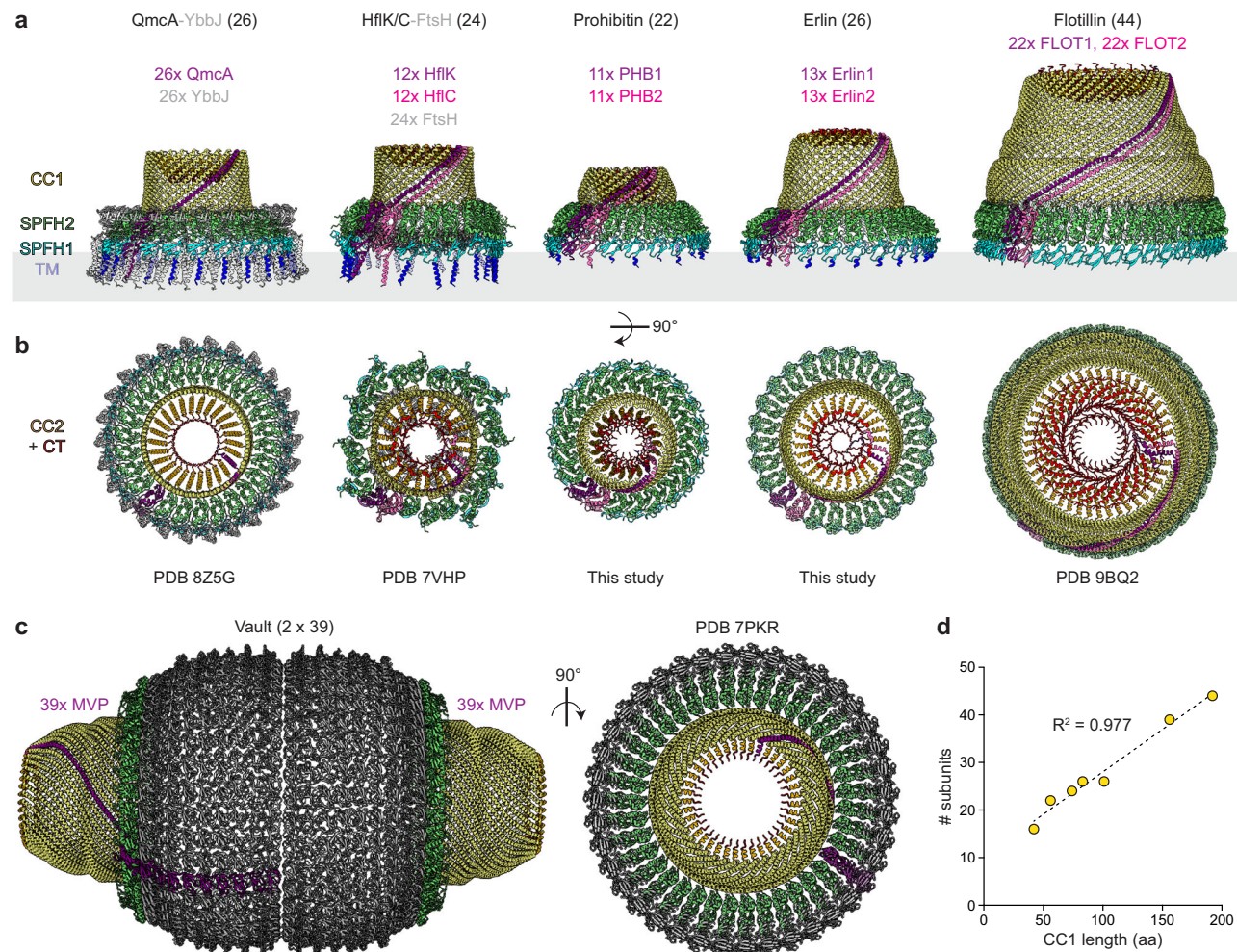

**Fig. 7 | Comparison of SPFH protein complexes. a** Side and **b** top views of the indicated membrane-associated SPFH protein complexes. **c** Side (left) and top (right) view of the major vault complex. Unique subunits are colored purple and pink (for SPFH proteins) or gray (for other associated proteins). All other copies of SPFH proteins in the complex are colored according to domain as indicated. The total number of SPFH protein subunits (in parentheses) and the relative stoichiometry of proteins (colored) in each complex are listed. **d** Plot of the length of the CC1 helix (based on the number of amino acids, aa) versus the number of subunits observed in experimentally determined structures of SPFH complexes, fitted with a linear regression (y = 0.1782x + 10.22). Source data are provided as a Source Data file.

to impact lipid homeostasis and organelle morphologies[8,39,59–63]. In addition to the membrane association of the Erlin1/2 and PHB1/2 complexes through N-terminal transmembrane and hydrophobic SPFH1 domain interfaces (Figs. 2c and 6b), our structures highlight unique properties of the C-terminal domains that make up the narrow end of the domed structures. These properties include distinct, tightly packed conformations formed by alternating CT placements and a putative central 'channel' with hydrophobic properties (Figs. 3, 6f, and Supplementary Figs. 4e, f and 10). The observations that a detergent micelle may stably interact with the interior of the C-terminal PHB1/2 structure (Supplementary Fig. 10d) and that the C-terminal 'cap' of the FLOT1/2 complex can engage lipid vesicles[5] raise the intriguing possibility that these surfaces may impart specific functions to SPFH complexes. Our findings provide a model to accelerate mechanistic investigations into these possibilities.

## Methods
### Plasmids and antibodies
Plasmids with Erlin1, Erlin2, PHB1, and PHB2 cDNAs and lentiviral plasmids were gifts from Wade Harper's lab. Each cDNA was cloned into a pcDNA5 mammalian expression vector with an N-terminal Strep or C-terminal Flag tag, and PHB1 was cloned into a pHAGE vector with

an N-terminal Strep tag using Gibson assembly (NEB E611). Mutations were introduced using Phusion (NEB M0530) mutagenesis and T4 ligation (NEB M0202).

HRP-conjugated anti-FLAG M2 (Sigma A8592, 1:5,000), HRP-conjugated StrepTactin (Bio-rad 1610381, 1:5,000), anti-Strep II (Abcam ab76949), anti-Erlin1 (Invitrogen PA5-19152, 1;1,000), anti-Erlin2 (Abcam ab128924, 1:4,000), anti-PHB1 (Invitrogen PA5-27329, 1:1,000), anti-PHB2 (Invitrogen PA5-14133, 1:1,000), anti-AFG3L2 (Abcam ab68023, 1:500), anti-SPG7 (Novus NBP2-01860, 1:500), anti-MIRO2 (Abcam ab224089, 1:1,000), HRP-conjugated goat anti-rabbit IgG (Jackson ImmunoResearch 111-035-003, 1:5,000), HRP-conjugated goat anti-mouse IgG (Jackson ImmunoResearch 115-035-003, 1:5,000), and HRP-conjugated rabbit anti-goat IgG (R&D Systems HAF017, 1:1,000) antibodies were purchased.

### Cell culture and cell line generation
HEK293T (ATCC CRL-3216) and Flp-In 293 T-REx (Invitrogen R78007) cells were cultured in DMEM with high glucose, GlutaMAX, and sodium pyruvate (Gibco 10566016) supplemented with 10% fetal bovine serum (FBS) at 37 °C and 5% $CO_2$. Expi293F cells (Gibco A14527) were cultured in Expi293 medium (Gibco A1435101) at 37 °C and 8% $CO_2$ with shaking at 120 rpm. Expi293F cells stably expressing Strep-tagged PHB1 was

generated by lentiviral transduction. pHAGE-PHB1 was transfected into HEK293T cells together with helper plasmids (pHDM-VSVG, pHDM-MGPM2, pHDM-tat1B, and pRC-CMV-rev1B) using TransIT-293 (Mirus), according to the manufacturer's instructions, in a 6-cm plate, and the medium was changed after 24 h. Forty-eight hours after transfection, the medium containing lentivirus was collected and syringe filtered through a 0.45 μm filter. 500 μL lentivirus-containing medium and polybrene at a final concentration of 10 μg/mL was inoculated into 25 mL Expi293F cells at a density of ~3 million cells/mL. After 48 h, the infected Expi293F cells were placed under selection with 2 μg/mL puromycin until the cells reached >90% viability. To generate knock-out cell lines, HEK293T or Flp-In 293 T-REx were transfected with pX459 with target gRNA sequences identified using CHOPCHOP[64] (Erlin1: GCTTTACTAACTAGCCCCAG, Erlin2: GCTGTGCACAAGATA-GAAGA) using TransIT-293. 48 h after transfection, cells were placed under selection with 1 μg/mL puromycin (Gibco A11138-03) for 48 h. Clonal lines were isolated and validated using immunoblotting and genotyping. To generate rescue Flp-In 293 T-REx cells, Erlin1 or Erlin2 knockout cells were cotransfected with a 1:1 ratio of pOG44 and pcDNA5/FRT/TO containing C-terminally Flag-tagged Erlin1 or Erlin2, respectively, behind a CMVd1 promoter using TransIT-293. 48 h after transfection, cells were placed under selection with 2.5 μg/mL blas-ticidin (Gibco A11139-03) and 50 μg/mL hygromycin (Millipore Sigma 31282-04-9) for 2–3 weeks. Expression was induced with 100 ng/mL doxycycline for 24 h and validated by immunoblotting.

### Protein expression and purification

To purify the Erlin1/2 complex, pcDNA5 plasmids encoding N-terminally Strep-tagged Erlin1 and C-terminally Flag-tagged Erlin2 were transfected at a concentration of 1 μg/mL of each plasmid into 1.2 L Expi293F cells at a density of ~2.5 million/mL using 6 μg/mL polyethyleneimine (PEI-25K; Polysciences, 23966). The cells were fed 24 h after transfection with 3 mM sodium valproate (Sigma P4543) and 0.45% glucose (Sigma G8769) and harvested 72 h after transfection by centrifugation at $1000 \times g$ for 10 min. The cell pellet was washed twice in cold PBS, lysed by sonication in 50 mM HEPES/KOH, pH 7.4, 150 mM NaCl, 1 mM dithiothreitol (DTT) with 1× cOmplete protease inhibitor cocktail (PIC; Roche 1187358), and clarified by centrifugation at $1000 \times g$ for 5 min. Crude cellular membranes were then isolated by ultracentrifugation at $120,000 \times g$ for 1 hr at 4 °C in a 50.2 Ti rotor (Beckman Coulter), resuspended in 50 mM HEPES/KOH, pH 7.4, 150 mM NaCl, and solubilized by the addition of 1% (w/v) n-dodecyl-β-D-maltoside (DDM, Anatrace D310) and 0.2% (w/v) cholesteryl hemi-succinate (CHS, Sigma C6512) at 4 °C for 2 h. The supernatant after ultracentrifugation at $100,000 \times g$ for 30 min at 4 °C in a 50.2 Ti rotor (Beckman Coulter) was incubated with StrepTactin HP resin (Cytiva, 28-9355) for 1.5 h at 4 °C, washed 3 times with wash buffer [50 mM HEPES/KOH, pH 7.4, 150 mM NaCl, 0.02% (w/v) DDM, 0.004% (w/v) CHS] and eluted with 3 column volumes of 10 mM desthiobiotin (Sigma D1411) in wash buffer. The elution was then incubated with anti-FLAG M2 resin (Sigma A2220) for 1.5 h at 4 °C and washed as above. Elutions were carried out with 3 column volumes of wash buffer plus 0.15 mg/mL 3×FLAG peptide (APEXBIO TECH MS MSPP-A6001). The eluate was concentrated to 500 μL and separated on a Superose 6 Increase 10/300 GL column (Cytiva 29091596) pre-equilibrated in wash buffer. Peak fractions were pooled and immediately plunge fro-zen for cryo-EM analysis (see below).

Purification of the PHB1/2 complex was adapted from a previous study[7]. Briefly, 1 μg/mL of a pcDNA5 plasmid encoding untagged PHB2 was transfected into 600 mL Expi293F cells stably expressing Strep-tagged PHB1 at a density of ~2.5 million/mL using 3 μg/mL PEI. The cells were fed 24 h after transfection with 3 mM sodium valproate and 0.45% glucose and harvested 48 h after transfection by centrifugation at $1000 \times g$ for 10 min. The cell pellet was washed with cold PBS, resus-pended, and lysed by sonication in lysis buffer (20 mM HEPES/KOH, pH

7.4, 50 mM K-phosphate, 150 mM KCl, 5 mM magnesium acetate, 5 mM ATP, 10% glycerol) with 1x PIC. After clarification by centrifugation at $1000 \times g$ for 5 min at 4 °C twice, crude membranes were isolated by centrifugation at $17,000 \times g$ for 15 min at 4 °C and solubilized in lysis buffer plus 2% (w/v) DDM and 0.4% (w/v) CHS at 4 °C for 30 min. The supernatant after centrifugation at $21,000 \times g$ for 15 min at 4 °C was incubated with StrepTactin HP (Cytiva, 28-9355) resin at 4 °C for 1.5 h. The resin was washed in wash buffer 1 [lysis buffer with 0.02% (w/v) DDM and 0.004% (w/v) CHS] and then in wash buffer 2 (the same as wash buffer 1 but with the glycerol concentration reduced to 1%). Elutions were carried out with 10 mM desthiobiotin in wash buffer 2 in ~100 μL fractions. The fraction with the highest protein concentration was directly plunge frozen for cryo-EM analysis.

### Cryo-EM sample preparation and data collection

A total of 3.5 μl of the Erlin1/2 complex sample, purified as described above, was applied to glow-discharged R1.2/1.3 Cu 300 mesh grids (Quantifoil) and frozen in liquid ethane using a Vitrobot Mark IV (Thermo Fisher Scientific) set at 4 °C and 100% humidity with a 10 sec wait time, 3 sec blot time, and +8 blot force. Grids of the PHB1/2 complex were prepared in the same way but with an additional wash step using buffer without glycerol before blotting.

The Erlin1/2 complex dataset was collected using a Titan Krios (Thermo Fisher Scientific) operating at 300 kV and equipped with a BioQuantum K3 imaging filter with a 20-eV slit width and a K3 summit direct electron detector (Gatan) in counting mode at a nominal mag-nification of 105,000× corresponding to a calibrated pixel size of 0.83 Å. Semi-automated data collection was performed with SerialEM v4.0.5. A 2.684 sec exposure was fractionated into 48 frames, resulting in a total exposure of 52.32 electrons per Å$^2$. The defocus targets were −1.2 to −2.2 μm.

The PHB1/2 complex dataset was collected using a Titan Krios (Thermo Fisher Scientific) operating at 300 kV and equipped with a Falcon 4i camera (Thermo Fisher Scientific) in counting mode at a nominal magnification of 165,000× corresponding to a calibrated pixel size of 0.73 Å. Semi-automated data collection was performed with EPU software (Thermo Fisher Scientific) and a total exposure time of 2.52 sec, corresponding to a total dose of 50 electrons per Å$^2$ and 774 EER movie frames. The defocus targets were −0.8 to −2.0 μm.

### Image processing and model building

For the Erlin1/2 complex dataset, 7391 micrographs were collected and subjected to auto-picking with templates generated from man-ual picking after motion correction and CTF estimation in RELION-4.0.1[65]. The particles were extracted with a box size of 512 and downsampled to a box size of 128 for initial classification steps. After 2D classification, 708,455 particles were selected for 3D classification with a low-resolution input reference from a previous dataset col-lected on a complex of Strep-tagged Erlin1 purified with coexpres-sion of untagged Erlin2. 129,017 particles after the initial 3D classification step were extracted in a box size of 256 and subjected to 3D refinement and two rounds of 3D classification with C13 symmetry applied. 37,658 particles selected after the 3D refine-ments were unbinned and subjected to CTF refinement and Bayesian polishing, followed by refinement with C26 symmetry applied. Par-ticles after C26 symmetry expansion (979,108) were subjected to focused 3D classification in cryoSPARC v4.3.1[66] with a mask on the C-terminal region of the complex. 95,469 particles in a class in which the CT backbones could be traced were used for local refinement without applied symmetry.

For the PHB1/2 complex dataset, 12,877 micrographs were col-lected, and data processing was performed in cryoSPARC v4.3.1. After patch-based motion correction and CTF estimation, micrographs with severe contamination or poor CTF fits were removed. 12,532 micro-graphs were subjected to automated particle picking using templates

generated from manual picking. The particles were extracted with a box size of 512 and downsampled to a box size of 128 for initial classification steps. After several rounds of 2D classification, 109,090 particles were selected for heterogeneous refinement using multiple reference volumes generated by ab initio reconstruction with C11 symmetry. 34,221 particles from the best class were unbinned and subjected to non-uniform and additional heterogeneous refinements. 22,299 particles corresponding to the closed conformation and 7010 particles corresponding to the open conformation were independently subjected to local CTF and non-uniform refinements.

## Model building and analysis

Unsharpened and postprocessed maps from cryoSPARC and deepEMhancer (20220530_cu10)[67] were used for interpretation. ModelAngelo v1.0.14[68] was used to generate initial models of the Erlin1/2 complex and of the PHB1/2 complex in the closed conformation. The models were then manually adjusted in Coot v.0.9.8[69], including the addition of glycans to the Erlin1/2 complex, in between iterative rounds of real space refinement in Phenix v.1.19.2[70]. The model of the closed PHB1/2 complex was then used as an initial model for the PHB1/2 complex in the open conformation, which was adjusted and refined in the same way. Cryo-EM data processing and model-building programs were supported by SBGrid[71]. Figures were made using ChimeraX v.1.8[72]. Sequence alignments were performed using Clustal Omega[73] and depicted using ESPript 3.0[74].

## Mutant complex analyses

For BN-PAGE analyses, wildtype, Erlin1 knockout (KO), Erlin2 KO, or Erlin1/2 double KO HEK293T or Flp-In 293 T-REx cells were plated in 6-well plates and harvested 48 h after. To analyze mutant Erlin complexes, Erlin1 or Erlin2 KO HEK293T cells were plated in 6-well plates and transfected with 2.5 μg of pcDNA5/FRT/TO plasmids encoding untagged or C-terminally Flag-tagged Erlin1 or Erlin2, respectively, using TransIT-293. To analyze mutant PHB complexes, Expi293F cells or wildtype HEK293T cells plated in 6-well plates were transfected with pcDNA/FRT/TO plasmids encoding N-terminally Strep-tagged PHB1 and C-terminally Flag-tagged PHB2 variants in a 1:1 ratio at a concentration of 2 μg plasmids per 1 mL of Expi293F cells using PEI as above, or 2.5 μg plasmids in each well of HEK293T cells using TransIT. Expi293F cells were harvested 48 hr after transfection. HEK293T cells were expanded to a 10 cm dish 24 h after transfection and harvested 48 h later.

All cells were harvested, pelleted, and washed twice in cold PBS, and then lysed in three cell pellet volumes of either 50 mM HEPES-KOH, pH 7.5, 100 mM KOAc, 2 mM Mg(OAc)$_2$, 1% (w/v) DDM, 1× PIC for Erlin-expressing cells, or 50 mM HEPES-KOH pH 7.5, 150 mM KCl, 5 mM Mg(OAc)$_2$, 10% (w/v) glycerol, 1% (w/v) DDM, 1× PIC for PHB-expressing cells by rotation for 30 min at 4 °C. The lysates were clarified by centrifugation at $21,000 \times g$ for 10 min at 4 °C. The concentrations of the supernatants were normalized based on absorbance at 280 nm, and 10-20 μg of the clarified lysates were loaded for 3–12% Bis-Tris Blue NativePAGE (Invitrogen BN1003) at 150 V for 115 min with dark blue cathode buffer for 30 min and light blue cathode buffer for the remaining time. The gels were rinsed with water and then soaked in transfer buffer (25 mM Tris, 200 mM glycine, 0.04% SDS) for 15 min before wet transfers onto 0.45 μm PVDF transfer membranes that were activated in methanol for 16 h at 30 V at 4 °C. The blots were rinsed in 40% methanol and 10% acetic acid for 10 min and then in 20% methanol in 25 mM Tris, 200 mM glycine for 10 min, before Ponceau staining and immunoblotting according to standard procedures. SDS-PAGE were performed on the same samples using 10% Tris-tricine gels, which were subjected to wet transfers in 25 mM Tris, 200 mM glycine, 20% MeOH, and 1% SDS onto 0.2 μm nitrocellulose membranes at 100 V for 50 min, followed by immunoblotting according to standard procedures.

## Reporting summary

Further information on research design is available in the Nature Portfolio Reporting Summary linked to this article.

## Data availability

The cryo-EM maps generated in this study have been deposited in the Electron Microscopy Data Bank (EMDB) under accession codes EMD-70263 (the Erlin1/2 complex), EMD-70267 (the closed PHB1/2 complex), and EMD-70268 (the open PHB1/2 complex). The atomic coordinates have been deposited in the Protein Data Bank (PDB) under accession codes PDB 9O9U (the Erlin1/2 complex), PDB 9O9Z (the closed PHB1/2 complex), and PDB 9OA0 (the open PHB1/2 complex). All other data are available within the article and its Supplementary Information. Source data are provided with this paper. Correspondence and requests for materials should be directed to S.S. Source data are provided with this paper.

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

## Acknowledgements

We thank Yinong Liu and Michael Rale for help with data analysis programs, and all Shao lab members for useful discussions. Cryo-EM screening and data collection were performed at the Harvard Center for Cryo-Electron Microscopy (HC2EM). Data processing was supported by SBGrid and Research Computing at Harvard Medical School. This work was supported by the Packard Foundation (S.S.), NIH R01 AG073277 (S.S.), a Smith Family Foundation Odyssey Award (S.S.), a Robin Reed Memorial Fellowship (J.G.), and an EMBO Long-Term Fellowship (D.S.). S.S. is an Investigator with the Howard Hughes Medical Institute.

## Author contributions

J.G. and S.S. conceived the project. J.G. purified all complexes, prepared cryo-EM samples, collected cryo-EM data, and performed the cryo-EM data processing and modeling with help from D.S. and N.K. J.G. and S.S. biochemically analyzed complex variants with help from D.S., H.C. and J.Z. S.S. supervised the project. J.G., D.S. and S.S. wrote the paper with input from all authors.

## Competing interests

The authors declare no competing interests.
