## [Transparent Peer Review file · Nature Communications]

Structures of human organellar SPFH protein complexes

Corresponding Author: Dr Sichen Shao

Version 0:

Reviewer comments:

Reviewer #1

(Remarks to the Author)

In this manuscript, Gao et al. provide the first detailed structural insights into two human organellar SPFH complexes, the Erlin1/2 and PHB1/2 complexes. Using cryo-EM, Gao et al. reveal that both complexes assemble into cage-like structures, comprising 26 subunits for the Erlin1/2 complex and 22 subunits for the PHB1/2 complex. The detailed structural analysis of these complexes offers some insights into how disruptions in their function could contribute to diseases such as hereditary spastic paraplegia, ALS, and mitochondrial disorders. In the context of previously published structural studies of prokaryotic SPFH complexes, this manuscript introduces novel findings, particularly regarding the conformational heterogeneity observed in the PHB1/2 complex. Furthermore, the first high-resolution structures of human SPFH complexes in this study provide a structural basis for understanding disease-associated mutations.

Overall, this manuscript represents a significant advancement in the field of structural biology by presenting the high-resolution cryo-EM structures of the human Erlin1/2 and PHB1/2 complexes. The identification of conformational heterogeneity, the mapping of disease-associated mutations, and the comparative analysis with prokaryotic SPFH family members offer novel insights into the function and evolution of these complexes. However, there are several concerns that need to be addressed before the manuscript can be considered for publication in Nature Communications.

Major points:

1. One of my major concerns with this manuscript is its organization. The paper presents the structures of two distinct complexes, but the connections between these two structural discoveries are weak or underrepresented. As it stands, the manuscript reads more like two separate studies joined together, rather than a cohesive, integrated piece of work. I strongly encourage the authors to elaborate on the relationships between their findings and provide a more seamless narrative that integrates both complexes into a unified framework.
2. Another major concern in reviewing this manuscript is the lack of functional validation for the key structural interactions that are described. The authors provide excellent high-resolution cryo-EM maps of the Erlin1/2 and PHB1/2 complexes, revealing important structural insights into the inter-subunit interactions, but the manuscript does not include direct experimental evidence demonstrating that these interactions are functionally relevant. Given that many of the conclusions rely on these structural interactions, I strongly recommend that the authors perform additional experiments to validate these interfaces and their roles in complex formation and function. Specifically, the authors should consider using site-directed mutagenesis to disrupt key residues at the interfaces, followed by biochemical assays to examine the impact on complex assembly and stability. This could be followed by functional assays, such as measuring ERAD efficiency for Erlin1/2 and mitochondrial protein quality control for PHB1/2, to confirm that these interactions are necessary for cellular function. Additionally, the authors could utilize techniques such as co-immunoprecipitation or pull-down assays to confirm the physical disruption of these interfaces in mutant forms of the proteins. Incorporating these types of functional and biochemical validations would greatly enhance the manuscript by demonstrating that the described structural features are not simply interesting observations but are essential for the activity and function of these protein complexes.
3. An interesting and potentially significant structural observation in this study is the conserved architecture of the C-terminal regions of the SPFH domains. The authors propose that the subunit organization in the Erlin1/2 complex follows a mixed-mode arrangement, with the Erlin1/2 ratio potentially varying across different samples. However, the narrow end of the Erlin1/2 cage suggests that only half of the complex's subunits could accommodate their C-terminal domains (CTDs) within this region. Given the divergent CTD residues between Erlin1 and Erlin2 (SNerlin1 vs KDerlin2), do these differences imply that only Erlin2 can extend its CTD into the narrow end? Considering that SPFH family proteins typically organize as heterodimers with substantial asymmetry at their C-termini—one helix projecting inward and the other outward—this observation might provide important insight into the assembly mechanism of the Erlin1/2 complex. Furthermore, the density presented in Supplementary Fig. 4 appears sufficient to distinguish between Erlin1 and Erlin2 subunits. Therefore, I strongly recommend that the authors consider reprocessing their cryo-EM data using C13 symmetry,

rather than the C26 symmetry currently employed. Such an approach could potentially improve the overall resolution, enhance the differentiation between the subunits, and thus facilitate more accurate subunit assignment, clarifying the heterodimeric organization of the complex. Additionally, it would be valuable to expand the analysis and discussion regarding the conservation and potential functional relevance of the C-terminal regions. A deeper exploration into their structural or regulatory roles could greatly enhance the mechanistic understanding of SPFH-family complex assembly and function.

4. The authors mention a potential interaction between the PHB1/2 complex and AAA+ proteases, such as AFG3L2. While the provided WB results suggest a potential interaction with AFG3L2, the current data do not clearly define a binding site or offer direct evidence for this association. To strengthen this claim, it would be beneficial to further explore these interactions using complementary techniques such as crosslinking mass spectrometry or proximity-labeling approaches, which can help detect transient or spatially proximal interactions in mitochondria. Additionally, considering the structural similarity between ctPHB-m-AAA protease assemblies and the observed architecture in this study, it would be useful to discuss whether the micellar density located beneath the PHB1/2 complex could represent a potential docking site for proteases or other interacting partners.

Minor points

1. In Supplementary Figure 5, the authors should include the residue numbers for the key residues shown in the graph to provide additional clarity and facilitate understanding of the structural data.
2. While Figure 6 provides a compelling overview of the diversity in SPFH protein complex architectures, including subunit stoichiometries and shapes, the relationship between the SPFH2 domain geometry and subunit copy number remains qualitative. The authors state that the subunit number generally correlates with the diameter of the wider end of the domed structure, which is primarily determined by the conserved dimensions of the SPFH2 domains. However, this key structural principle could be made clearer by including a more explicit comparison—such as estimating the angular span occupied by each SPFH2 domain in different assemblies, or correlating domain length with measured complex diameter. Incorporating such analysis would not only support the stated hypothesis but also help the reader understand how SPFH domain geometry constrains or determines the overall architecture and stoichiometry of these complexes.
3. In the last two paragraphs of the section “Key interactions within the Erlin1/2 complex,” the authors use Figure 3e to illustrate the interaction of the Erlin CTDs with their surrounding hydrophobic environment. However, Figure 3e suffers from overlapping side chains and residue labels, which may confuse readers. I suggest the authors revise Figure 3e by splitting it into two panels, each showing the CTDs of Erlin1 and Erlin2 separately. Additionally, in the description, the authors refer to the interaction with “CC2 and β 9 of the heterodimer to the right, when viewed from the outside of the cage.” This phrasing may be unclear, as different readers could interpret “left/right” differently, even with the view angle specified as “the outside of the cage.” Using terms like “clockwise” or “counter-clockwise” might provide more clarity.
4. In the section “Positions of disease-linked Erlin1/2 mutations”, the authors only described the locations of the pathogenic mutations and provided some structural insight into the consequences of those mutations. Readers might benefit from a deep discussion on the biological impact of those mutations and how these impacts would cause disease.
5. In Supplementary Figure 2, the reported particle counts and their corresponding percentages in the processing workflow appear inconsistent; please verify and correct these values.

Reviewer #2

(Remarks to the Author)

Gao et al. present elegant cryo-EM structures of human organellar SPFH-family protein complexes, specifically Erlin1/2 (localized to the endoplasmic reticulum) and PHB1/2 (localized to the mitochondrial inner membrane). These SPFH proteins are key organizers of membrane microdomains and modulators of membrane protein function within their respective organelles.

Among the many important findings, the authors reveal that the Erlin1/2 complex assembles into a striking 26-subunit structure. Several disease-associated mutations in Erlin map to structurally critical regions of the complex, providing mechanistic insight into the pathogenesis of Erlin-related disorders.

The authors also resolve the structure of the PHB1/2 complex, which forms a dimer of 22-subunit rings—comprised of 11 PHB1/PHB2 heterodimers—resulting in a dome-like assembly. While previous studies have linked the PHB complex to mitochondrial matrix AAA protease regulation, the current structures do not reveal associated protease densities, suggesting that these interactions may be transient, sub-stoichiometric, or spatially segregated. Nonetheless, the structures of the PHB1/2 complexes offer substantial new insight into their potential scaffolding and regulatory functions in mitochondrial organization.

Overall, this is a well-executed and impactful study that significantly advances our understanding of SPFH-family protein architecture and function. I recommend publication in Nature Communications without further revisions.

Minor comments:

1. In the section describing the generation of the PHB1/2 complex for cryo-EM (Methods and Supplementary Fig. 6), the authors used an N-terminally Strep-tagged PHB1 construct stably expressed in Expi293F cells, followed by transient expression of untagged PHB2. I am curious whether the authors have performed localization studies—such as immunofluorescence—to confirm that the tagged PHB1 localizes correctly to mitochondria. It is important to rule out the possibility that the N-terminal tag interferes with mitochondrial import, potentially leading to mislocalization to the ER or plasma membrane. Even a brief comment on this point would be helpful to clarify this.
2. Given the high-resolution structures, were any lipid densities observed or identified? While lipidomics is beyond the scope of this study, it could be highly informative in future work to determine whether specific lipids—such as cardiolipin or cholesterol—are associated with these complexes. Even speculative comments on potential lipid interactions based on observed densities would provide a valuable reference point for advancing our understanding of how these complexes interact with their membrane environments.

Z.F.

Reviewer #3

(Remarks to the Author)

Version 1:

Reviewer comments:

Reviewer #1

(Remarks to the Author)

I have carefully reviewed the authors' responses to my previous comments and suggestions, as well as the revisions made to the manuscript. I find that the authors have satisfactorily addressed the concerns raised in the original review, and the revised version is substantially improved in both clarity and coherence. I therefore support the publication of this work in Nature Communications.

Reviewer #2

(Remarks to the Author)

All of my comments have been addressed in the revision.

Reviewer #3

(Remarks to the Author)

Reply to Reviewers

We thank the reviewers for their helpful feedback, which we have used to improve the revised manuscript. Together with point-by-point replies to their comments below, notable changes include:

1. Rearrangement of the Introduction to emphasize the conceptual links between the Erlin1/2 and PHB1/2 complexes compared to other SPFH complexes.
2. Addition of data analyzing Erlin1/2 and PHB1/2 complex assembly (new Supplementary Fig. 4a,b, new Supplementary Fig. 5, and new Supplementary Fig. 10e,f).
3. Addition of figure panels (new Supplementary Fig. 3f and new Supplementary Fig. 9d) showing additional densities associated with the complexes, and
4. Changes to some figures (e.g., Fig. 3e and Supplementary Fig. 2) to improve interpretation.

Reviewer 1

In this manuscript, Gao et al. provide the first detailed structural insights into two human organellar SPFH complexes, the Erlin1/2 and PHB1/2 complexes. Using cryo-EM, Gao et al. reveal that both complexes assemble into cage-like structures, comprising 26 subunits for the Erlin1/2 complex and 22 subunits for the PHB1/2 complex. The detailed structural analysis of these complexes offers some insights into how disruptions in their function could contribute to diseases such as hereditary spastic paraplegia, ALS, and mitochondrial disorders. In the context of previously published structural studies of prokaryotic SPFH complexes, this manuscript introduces novel findings, particularly regarding the conformational heterogeneity observed in the PHB1/2 complex. Furthermore, the first high-resolution structures of human SPFH complexes in this study provide a structural basis for understanding disease-associated mutations.

Overall, this manuscript represents a significant advancement in the field of structural biology by presenting the high-resolution cryo-EM structures of the human Erlin1/2 and PHB1/2 complexes. The identification of conformational heterogeneity, the mapping of disease-associated mutations, and the comparative analysis with prokaryotic SPFH family members offer novel insights into the function and evolution of these complexes. However, there are several concerns that need to be addressed before the manuscript can be considered for publication in Nature Communications.

We thank the reviewer for pointing out the value of the structural insights into these complexes and include responses to the specific points raised below.

Major points:

1. One of my major concerns with this manuscript is its organization. The paper presents the structures of two distinct complexes, but the connections between these two structural discoveries are weak or underrepresented. As it stands, the manuscript reads more like two separate studies joined together, rather than a cohesive, integrated piece of work. I strongly encourage the authors to elaborate on the relationships between their findings and provide a more seamless narrative that integrates both complexes into a unified framework.

We thank the reviewer for the feedback and edited the manuscript to improve continuity by highlighting the specific similarities between the Erlin and PHB complexes relative to other SPFH complexes in the Introduction. We note this is primarily a stylistic issue and summarize two considerations that are part of our reasoning for presenting the structures of both complexes in one manuscript. First, conceptually, there are distinct similarities regarding the *putative* functions of the Erlin and PHB complexes that are most analogous to the prokaryotic HflK/C complex but not shared with most other eukaryotic SPFH complexes. We now emphasize these similarities in the Introduction. Second, although hypothetical function(s) are assigned to the Erlin and PHB complexes, it is important to emphasize that **their molecular mechanisms remain incompletely defined**. Notably, there are no well-established, robust assays for a specific function of either complex reproduced across different labs—or any indication if readouts are direct vs. indirect outcomes of perturbing either complex. In this context, with the data we

have on hand, we believe it is more rigorous to present a study comparing the experimentally determined structures rather than speculating on their individual (and still-debated) functions.

2. Another major concern in reviewing this manuscript is the lack of functional validation for the key structural interactions that are described. The authors provide excellent high-resolution cryo-EM maps of the Erlin1/2 and PHB1/2 complexes, revealing important structural insights into the inter-subunit interactions, but the manuscript does not include direct experimental evidence demonstrating that these interactions are functionally relevant. Given that many of the conclusions rely on these structural interactions, I strongly recommend that the authors perform additional experiments to validate these interfaces and their roles in complex formation and function. Specifically, the authors should consider using site-directed mutagenesis to disrupt key residues at the interfaces, followed by biochemical assays to examine the impact on complex assembly and stability.

Several interfaces were previously analyzed for their impacts on protein stability and complex assembly, albeit not based on experimental structures. For example, PMID: 19131330 and 22020079 showed that mutations in the Erlin2 CT impaired complex assembly, and PMID: 38782601, 30135210, 38607533, and 37752894 looked at disease-linked Erlin2 mutations (R36K, G48V, H50Y, T65I, V186M, V71A). We also added data showing the impact of structure-based mutagenesis on complex assembly and stability for Erlin1/2 (new Supplementary Fig. 5) and PHB1/2 (new Supplementary Fig. 10e,f).

This could be followed by functional assays, such as measuring ERAD efficiency for Erlin1/2 and mitochondrial protein quality control for PHB1/2, to confirm that these interactions are necessary for cellular function. Additionally, the authors could utilize techniques such as co-immunoprecipitation or pull-down assays to confirm the physical disruption of these interfaces in mutant forms of the proteins. Incorporating these types of functional and biochemical validations would greatly enhance the manuscript by demonstrating that the described structural features are not simply interesting observations but are essential for the activity and function of these protein complexes.

We agree that informative functional analyses of mutants would be nice. However, as mentioned above, the field still lacks a specific consensus function for either complex. While there is a strong conceptual analogy between the Erlin1/2 and PHB1/2 complexes and the better-defined prokaryotic HflK/C complex, identifying and developing a reproducible and direct functional readout would be beyond the scope of (and space available in) this study. The functional impacts associated with these complexes are primarily descriptive, which may reflect indirect phenotypes, or focus on individual reporters that may not be broadly representative. Meanwhile, unpublished proteomics analysis of wildtype, Erlin1 knockout (and rescue), Erlin2 knockout (and rescue), and Erlin1&2 double knockout cells do not show clear changes in the levels of inositol 1,4,5-triphosphate receptors (ITPR1, ITPR2, or ITPR3), the most commonly cited Erlin-dependent ERAD substrate (PMID: 17502376, 19240031, 30135210, 35567199; although we did not activate the receptors, which enhances their degradation), nor of RNF170 or TMUB1, which have been proposed to interact and function with the Erlin1/2 complex (PMID: 19240031, 21610068, 38782601) (Reviewer Fig. 1). Our unbiased datasets resemble other 'omics datasets that point towards potential trends, but not a unifying function (PMID: 38782601). Perturbations of the PHB complex have even more pleiotropic reported outcomes (e.g., PMID: 23144624, 23863811, 10207067, 10835343). Importantly, to definitively study the function of these complexes – particularly for detailed mutant analyses, we believe it will be imperative to investigate the proteins at near-endogenous expression levels. Although overexpression systems produce informative biochemical readouts regarding complex assembly, it is not yet clear how different copy numbers of these proteins – or how they are tagged – may impact endogenous functional interactions or mechanisms. Proceeding with overexpression systems and poorly defined functional assays, which are the tools that we currently have on hand, would be premature and may lead to problematic results and interpretations. On the other hand, establishing the appropriate experimental systems by performing endogenous tagging and mutagenesis, or by carefully titrating the expression of stably integrated constructs in knockout cells, would require more time and resources than are practical for revisions.

Reviewer Fig. 1. Quantitative proteomics of Erlin mutant cells. For each protein indicated in the bottom left of each box, the Log₂ fold-change (FC) in the levels of that protein in Erlin1 knockout (E1KO), Erlin2 knockout (E2KO), or Erlin1 and Erlin2 double knockout (DKO) cells relative to wildtype (WT), Erlin1 rescue (1-resc.) or Erlin2 rescue (2-resc.) cells are plotted. Relative protein levels in the indicated Flp-In 293 T-REx cells were obtained using multiplexed quantitative proteomics via tandem mass tagging (TMT-MS). Rescue cells were generated by integrating the indicated Erlin protein into the Flp-In locus behind a doxycycline-inducible promoter of Erlin1 KO (for 1-resc.) or Erlin2 KO (for 2-resc.) cells. All cells were induced with 100 ng/mL doxycycline for 24 hr before processing. Note: although the levels of Erlin1 and Erlin2 change as expected in the knockout and rescue cells (left), this has minimal impact on the levels of other proteins linked to the complex.

3. An interesting and potentially significant structural observation in this study is the conserved architecture of the C-terminal regions of the SPFH domains. The authors propose that the subunit organization in the Erlin1/2 complex follows a mixed-mode arrangement, with the Erlin1/2 ratio potentially varying across different samples. However, the narrow end of the Erlin1/2 cage suggests that only half of the complex's subunits could accommodate their C-terminal domains (CTDs) within this region. Given the divergent CTD residues between Erlin1 and Erlin2 (SNerlin1 vs KDerlin2), do these differences imply that only Erlin2 can extend its CTD into the narrow end? Considering that SPFH family proteins typically organize as heterodimers with substantial asymmetry at their C-termini—one helix projecting inward and the other outward—this observation might provide important insight into the assembly mechanism of the Erlin1/2 complex.

Indeed, we and a concurrent structural study (Yan et al. *bioRxiv* doi: 10.1101/2025.06.14.659634) both propose a primary architecture for the Erlin1/2 complex comprising thirteen Erlin1/2 heterodimers arranged such that the Erlin1 CTs extend inward and the Erlin2 CTs each project a helix outward. There are many reasons to favor this interpretation. However, we do not believe that the data in either study can conclusively rule out possible variations to this arrangement, especially when considering how heterogeneity may be handled and averaged during cryo-EM processing. We think it is important to note the possibility of such variations in the text instead of claiming an invariant architecture.

Important considerations regarding this point include:

- The residues that are distinct between the two paralogs (e.g., S301 and N302 in Erlin1 and K299 and D300 in Erlin2) are not conserved across different model organisms (new Supplementary Fig. 5f,g). If these residues were critical for dictating a specific structure, we might expect that the amino acid identities would be conserved with each paralog over evolution.
- The observation that some organisms, such as *C. elegans* only have one Erlin protein (PMID: 22269071), and that others, such as *Drosophila*, appear to completely lack Erlin proteins, further suggest that the architecture of the complex may not be so rigidly defined.
- Our observations (new Supplementary Fig. 4a,b and 5) and previous studies (e.g., PMID: 30135210, 19131330), indicate that Erlin1 and Erlin2 can independently oligomerize in the absence of the other, and that the stability of one paralog does not strictly depend on the other. Considered together with the particularly high sequence similarity between the two paralogs and the lack of specific sequence conservation across evolution, it would be extremely hard to rule out the possibility of alternative subunit arrangements.
- Finally, based on alanine mutations, distinct residues noted above are important for complex assembly; however, swapping the residues to that of the other paralog (e.g., SN Erlin1 + mutated K299S / D300N Erlin2 or mutated S301K / N302D Erlin1 + KD Erlin2) does not disrupt complex assembly (new Supplementary Fig. 5). This again suggests that a specific interaction between these distinct residues in the two paralogs is not critical.

Furthermore, the density presented in Supplementary Fig. 4 appears sufficient to distinguish between Erlin1 and Erlin2 subunits. Therefore, I strongly recommend that the authors consider reprocessing their cryo-EM data using C13 symmetry, rather than the C26 symmetry currently employed. Such an approach could potentially improve the overall resolution, enhance the differentiation between the subunits, and thus facilitate more accurate subunit assignment, clarifying the heterodimeric organization of the complex.

Conclusively distinguishing between the subtle sequence differences of the two paralogs would require local resolutions of ~ 2.5 Å and, importantly: homogenous complexes. In response to this reviewer's Minor Point 5, we have updated Supplementary Fig. 2 with intermediate 3D classification steps, which we apologize for omitting previously. In these steps, we did process the complex using C13 symmetry but could not get to high enough resolutions to unambiguously distinguish between Erlin1 and Erlin2, especially with the additional consideration that applying symmetry may introduce artefacts around the axis of symmetry, near the divergent CTs. Because of this, we performed the refinement with C26 symmetry – **but ONLY to perform a C26 particle expansion**. We did not use the C26 refinement for any direct interpretations. We reasoned that the C26 expansion (which essentially makes a copy of every subunit in each particle) would be our best bet to distinguish Erlin1 and Erlin2. However, although we tried different masks for signal subtraction and classifications without alignments using the C26-expanded particles, they did not definitively differentiate between Erlin1 and Erlin2. Finally, we did not want to apply additional symmetry to downstream refinements using subsets of the symmetry-expanded particles, as this would lead to duplicated particles.

Additionally, it would be valuable to expand the analysis and discussion regarding the conservation and potential functional relevance of the C-terminal regions. A deeper exploration into their structural or regulatory roles could greatly enhance the mechanistic understanding of SPFH-family complex assembly and function.

The Erlin CTs are the most divergent regions of the proteins, as we already state in the text, and are not well-conserved between paralogs across species (new Supplementary Fig. 5f,g). We added experimental analysis and discussion regarding these points. However, because we only have the structure of the human complex, and the narrow end of the complex is more prone to artefacts induced from refinements with applied symmetry, we do not feel comfortable making additional speculations.

4. The authors mention a potential interaction between the PHB1/2 complex and AAA+ proteases, such as AFG3L2. While the provided WB results suggest a potential interaction with AFG3L2, the current

data do not clearly define a binding site or offer direct evidence for this association. To strengthen this claim, it would be beneficial to further explore these interactions using complementary techniques such as crosslinking mass spectrometry or proximity-labeling approaches, which can help detect transient or spatially proximal interactions in mitochondria. Additionally, considering the structural similarity between ctPHB–m-AAA protease assemblies and the observed architecture in this study, it would be useful to discuss whether the micellar density located beneath the PHB1/2 complex could represent a potential docking site for proteases or other interacting partners.

The proposed interaction between the PHB1/2 complex and mitochondrial AAA proteases was reported by others (PMID: 10207067, 39657011). We agree that our data do not offer strong supporting evidence for this interaction and may actually instead indicate that the interaction is transient or indirect, as we state in the text. Thus, **we are not claiming that these factors directly interact in this study.** Existing crosslinking-mass spectrometry datasets (e.g., PMID: 38632225, 29222160) indeed report crosslinks between PHB2 and mitochondrial AAA proteases (AFG3L2 and SPG7) among crosslinks of over 50 other distinct proteins with the PHB proteins (Reviewer Fig. 2).

Name	Avg models	pDOCKQ	ipTM	Avg pLDDT	Avg PAE	Contacts	Name	Avg models	pDOCKQ	ipTM	Avg pLDDT	Avg PAE	Contacts
PHB1_HUMAN_PHB2_HUMAN_571aa	2.8	0.668	0.693	85.6	2.68	194	PHB2_HUMAN_MICU1_HUMAN_775aa	0	0.019	0.128	0	30	0
PHB1_HUMAN_MIX23_HUMAN_416aa	1.75	0.087	0.341	76.3	11.83	36	CH60_HUMAN_PHB2_HUMAN_872aa	0	0.033	0.127	0	30	0
TIM10_HUMAN_PHB2_HUMAN_389aa	1	0.179	0.165	56.6	14.75	36	PHB1_HUMAN_KAD2_HUMAN_511aa	0	0.045	0.125	0	30	0
COA6_HUMAN_PHB2_HUMAN_424aa	2.56	0.38	0.825	94.9	1.26	35	PAPS1_HUMAN_PHB1_HUMAN_896aa	0	0.56	0.125	0	30	0
TIM8A_HUMAN_PHB2_HUMAN_396aa	1.03	0.103	0.401	64.9	9.9	30	PHB1_HUMAN_MIC60_HUMAN_1030aa	0	0	0.125	0	30	0
PHB1_HUMAN_F136A_HUMAN_410aa	1.17	0.125	0.275	66.2	12.81	27	PPIA_HUMAN_PHB2_HUMAN_464aa	0	0.207	0.124	0	30	0
PHB1_HUMAN_CCHL_HUMAN_540aa	1.08	0.288	0.25	67.2	14.21	24	PHB2_HUMAN_OCAD1_HUMAN_544aa	0	0.21	0.124	0	30	0
CPT2_HUMAN_PHB2_HUMAN_957aa	1	0.335	0.347	72.2	11.87	23	SCO1_HUMAN_PHB2_HUMAN_600aa	0	0.022	0.124	0	30	0
MIX23_HUMAN_PHB2_HUMAN_443aa	1.04	0.056	0.316	75.3	12.43	14	PHB1_HUMAN_MTX1_HUMAN_738aa	0	0.019	0.124	0	30	0
PHB1_HUMAN_CHCH2_HUMAN_423aa	1	0.114	0.238	75.3	14.15	13	PHB1_HUMAN_EXOG_HUMAN_640aa	0	0.277	0.123	0	30	0
SDHA_HUMAN_PHB2_HUMAN_963aa	1	0.081	0.394	84.4	13.55	11	T117B_HUMAN_PHB2_HUMAN_471aa	0	0.208	0.122	0	30	0
PHB2_HUMAN_CQ10B_HUMAN_537aa	1	0.089	0.231	72.7	14.67	3	PHB1_HUMAN_TIM50_HUMAN_625aa	0	0	0.122	0	30	0
RMD1_HUMAN_PHB2_HUMAN_613aa	1	0.284	0.162	70.3	15	3	ZC3HF_HUMAN_PHB2_HUMAN_725aa	0	0.028	0.122	0	30	0
PHB1_HUMAN_COA6_HUMAN_397aa	1	0.034	0.14	65.8	15	2	COX19_HUMAN_PHB2_HUMAN_389aa	0	0.022	0.121	0	30	0
F162A_HUMAN_PHB2_HUMAN_453aa	1	0.122	0.117	66.3	15	2	PHB1_HUMAN_RANB3_HUMAN_839aa	0	0.021	0.121	0	30	0
PHB1_HUMAN_COX16_HUMAN_378aa	1	0.086	0.116	65.3	15	2	RAB7A_HUMAN_PHB2_HUMAN_506aa	0	0.055	0.114	0	30	0
F136A_HUMAN_PHB2_HUMAN_437aa	1	0.087	0.157	66.4	15	1	QCR8_HUMAN_PHB1_HUMAN_354aa	0	0.058	0.113	0	30	0
PHB1_HUMAN_OCAD1_HUMAN_517aa	1	0.1	0.12	65	15	1	TIM50_HUMAN_PHB2_HUMAN_652aa	0	0.019	0.113	0	30	0
PHB1_HUMAN_MYO5A_HUMAN_2127aa	0	0.027	0.213	0	30	0	PHB2_HUMAN_EXOG_HUMAN_667aa	0	0.565	0.113	0	30	0
SCO1_HUMAN_PHB1_HUMAN_573aa	0	0.046	0.187	0	30	0	MAIP1_HUMAN_PHB2_HUMAN_590aa	0	0.132	0.112	0	30	0
LETM1_HUMAN_PHB2_HUMAN_1038aa	0	0.063	0.175	0	30	0	CYC_HUMAN_PHB2_HUMAN_404aa	0	0.018	0.109	0	30	0
PHB2_HUMAN_SPG7_HUMAN_1094aa	0	0.019	0.173	0	30	0	CY1_HUMAN_PHB2_HUMAN_624aa	0	0.301	0.108	0	30	0
IMB1_HUMAN_PHB2_HUMAN_1175aa	0	0.592	0.17	0	30	0	PHB2_HUMAN_PGES2_HUMAN_676aa	0	0.58	0.108	0	30	0
ATD3A_HUMAN_PHB2_HUMAN_933aa	0	0.115	0.167	0	30	0	KCRU_HUMAN_PHB1_HUMAN_689aa	0	0.051	0.108	0	30	0
ECHA_HUMAN_PHB2_HUMAN_1062aa	0	0.024	0.165	0	30	0	TIM44_HUMAN_PHB2_HUMAN_751aa	0	0	0.108	0	30	0
KIF4A_HUMAN_PHB2_HUMAN_1531aa	0	0.219	0.165	0	30	0	PHB2_HUMAN_TIM21_HUMAN_547aa	0	0.029	0.107	0	30	0
AIFM1_HUMAN_PHB1_HUMAN_885aa	0	0.246	0.164	0	30	0	PHB1_HUMAN_PA2G4_HUMAN_666aa	0	0.101	0.107	0	30	0
PHB2_HUMAN_AFG32_HUMAN_1096aa	0	0	0.164	0	30	0	PHB1_HUMAN_HEM6_HUMAN_726aa	0	0.467	0.107	0	30	0
QCR2_HUMAN_PHB2_HUMAN_752aa	0	0.024	0.159	0	30	0	PHB2_HUMAN_OPA3_HUMAN_478aa	0	0.205	0.106	0	30	0
AIFM1_HUMAN_PHB2_HUMAN_912aa	0	0.029	0.159	0	30	0	PHB2_HUMAN_COX11_HUMAN_575aa	0	0.022	0.104	0	30	0
YMEL1_HUMAN_PHB2_HUMAN_1072aa	0	0.184	0.158	0	30	0	SURF1_HUMAN_PHB2_HUMAN_599aa	0	0.151	0.102	0	30	0
PHB1_HUMAN_TIDC1_HUMAN_557aa	0	0.335	0.157	0	30	0	PHB1_HUMAN_COX17_HUMAN_335aa	0	0.26	0.101	0	30	0
UCRI_HUMAN_PHB2_HUMAN_573aa	0	0.041	0.157	0	30	0	NDUA8_HUMAN_PHB2_HUMAN_471aa	0	0.021	0.1	0	30	0
AGK_HUMAN_PHB2_HUMAN_721aa	0	0.118	0.157	0	30	0	PHB2_HUMAN_TIM16_HUMAN_424aa	0	0.027	0.098	0	30	0
PHB1_HUMAN_YMEL1_HUMAN_1045aa	0	0.047	0.156	0	30	0	PHB2_HUMAN_NDUF4_HUMAN_474aa	0	0.051	0.096	0	30	0
PHB2_HUMAN_TIDC1_HUMAN_584aa	0	0.032	0.154	0	30	0	PHB2_HUMAN_PFD2_HUMAN_453aa	0	0.111	0.095	0	30	0
PHB1_HUMAN_SCMC1_HUMAN_749aa	0	0.236	0.149	0	30	0	NEUM_HUMAN_PHB2_HUMAN_537aa	0	0	0.094	0	30	0
NDUC2_HUMAN_PHB2_HUMAN_418aa	0	0.35	0.148	0	30	0	MDHM_HUMAN_PHB2_HUMAN_637aa	0	0	0.094	0	30	0
SCMC1_HUMAN_PHB2_HUMAN_776aa	0	0.391	0.148	0	30	0	CCHL_HUMAN_PHB2_HUMAN_567aa	0	0.447	0.093	0	30	0
PHB2_HUMAN_S2513_HUMAN_974aa	0	0.167	0.146	0	30	0	PHB2_HUMAN_NIP5_HUMAN_583aa	0	0.027	0.092	0	30	0
NDUC2_HUMAN_PHB1_HUMAN_391aa	0	0.292	0.144	0	30	0	TIM14_HUMAN_PHB2_HUMAN_415aa	0	0.033	0.089	0	30	0
PHB1_HUMAN_CYC_HUMAN_377aa	0	0.025	0.139	0	30	0	KAD2_HUMAN_PHB2_HUMAN_538aa	0	0	0.089	0	30	0
PHB2_HUMAN_ACAD9_HUMAN_920aa	0	0.153	0.139	0	30	0	PHB2_HUMAN_TIM9_HUMAN_388aa	0	0.112	0.087	0	30	0
COX20_HUMAN_PHB2_HUMAN_417aa	0	0.432	0.134	0	30	0	C1QBP_HUMAN_PHB2_HUMAN_581aa	0	0.02	0.087	0	30	0
PHB1_HUMAN_OPA3_HUMAN_451aa	0	0.054	0.134	0	30	0	CRIP2_HUMAN_PHB2_HUMAN_507aa	0	0.019	0.086	0	30	0
CISY_HUMAN_PHB2_HUMAN_765aa	0	0.065	0.134	0	30	0	CX6B1_HUMAN_PHB2_HUMAN_385aa	0	0.057	0.083	0	30	0
MIC60_HUMAN_PHB2_HUMAN_1057aa	0	0	0.134	0	30	0	PHB1_HUMAN_TIM14_HUMAN_388aa	0	0.067	0.083	0	30	0
COX6C_HUMAN_PHB2_HUMAN_374aa	0	0.085	0.132	0	30	0	MIF_HUMAN_PHB1_HUMAN_387aa	0	0.025	0.081	0	30	0
EM55_HUMAN_PHB2_HUMAN_765aa	0	0.025	0.131	0	30	0	NDU55_HUMAN_PHB2_HUMAN_405aa	0	0	0.072	0	30	0
NDUA4_HUMAN_PHB2_HUMAN_380aa	0	0.169	0.129	0	30	0	MIF_HUMAN_PHB2_HUMAN_414aa	0	0	0.068	0	30	0

Reviewer Fig. 2. Crosslinking-mass spectrometry hits with PHB proteins. List of all crosslinking partners – 100 in total – identified with PHB1 or PHB2 in the crosslinking-mass spectrometry datasets generated by Zhu et al., (2024) PMID: 38632225, with mitochondrial AAA proteases highlighted. All targets were submitted to AlphaFold2 jobs and the resulting parameters as described in PMID: 40015271 are listed. Note: the mitochondrial AAA proteases are among dozens of putative PHB crosslinking partners.

Hence, even if these crosslinking data were used as constraints, the accuracy structural analyses would be questionable because both PHB1 and PHB2, as well as mitochondrial AAA proteases, oligomerize. Although we now have an experimentally-defined structure the PHB1/2 complex that will facilitate interpretations of complex interactions, it remains unclear if endogenous mitochondrial AAA

protease complexes comprise 6 copies of AFG3L2, as reported in a structure of recombinantly-expressed and purified truncated catalytic domains of AFG3L2 (PDB 6NYY, PMID: 31327635), 6 copies of SPG7, or mixtures of the two, as suggested in studies such as PMID: 14623864, 28396416. The possible sources of crosslinks with these potential variations in copy number therefore increase exponentially. At this stage, especially when considering the focus of our study is on the core organization of organellar SPFH complexes, we do not believe that additional speculation regarding these putative interactions would be helpful in the absence of more definitive data.

Minor points

1. In Supplementary Figure 5, the authors should include the residue numbers for the key residues shown in the graph to provide additional clarity and facilitate understanding of the structural data.

We added residue numbers to these panels (which are now Supplementary Fig. 4c,d).

2. While Figure 6 provides a compelling overview of the diversity in SPFH protein complex architectures, including subunit stoichiometries and shapes, the relationship between the SPFH2 domain geometry and subunit copy number remains qualitative. The authors state that the subunit number generally correlates with the diameter of the wider end of the domed structure, which is primarily determined by the conserved dimensions of the SPFH2 domains. However, this key structural principle could be made clearer by including a more explicit comparison—such as estimating the angular span occupied by each SPFH2 domain in different assemblies, or correlating domain length with measured complex diameter. Incorporating such analysis would not only support the stated hypothesis but also help the reader understand how SPFH domain geometry constrains or determines the overall architecture and stoichiometry of these complexes.

We thank the reviewer for the suggestion and agree that these comparisons are somewhat qualitative, reflecting the diversity of SPFH complexes. We first re-emphasize that *individual* SPFH2 domains have conserved dimensions because they have a conserved structural fold. We apologize if this was not clearer and have adjusted our wording. Aligning the isolated SPFH2 domain of *any* SPFH protein with that of any other results in RMSD values of $<1 - 5 \text{ \AA}$. Thus, because the SPFH2 domains always oligomerize together into a ring, it is not especially surprising that increasing subunit numbers correlate with larger complex dimensions. While this may seem somewhat trivial, we felt it was still worth noting along with the visual comparisons in Fig. 7. Accordingly, plotting the subunit number versus measured diameters of the SPFH2 rings reveals a reasonable correlation (Reviewer Fig. 3, $R^2 = 0.932$). However, this comparison is not particularly useful, because it requires knowing the subunit number of the complex to have predictive value, which still requires experimental structure determination. The dimensions of SPFH complexes are also impacted by additional protein domains (such as in the major vault protein) or accessory proteins (such as the association of YbbJ with QmcA). Instead, a more useful metric would facilitate the estimation of the subunit number in SPFH complexes with undetermined structures. Toward this end, we found that the number of amino acids in the CC1 helix of each SPFH protein correlates better with subunit number (new Fig. 7d and Reviewer Fig. 3; $R^2 = 0.977$), and the linear regression can estimate subunit numbers within 2 of the actual copy number.

Reviewer Fig. 3. SPFH complex correlations.

Plot of the number of subunits versus the diameter of the SPFH2 ring (left) or of the length of the CC1 helix, in amino acids (aa) versus the number of subunits (right) in SPFH complexes with experimentally determined structures. Linear regressions and R^2 values are shown for each. Note: better correlation between the CC1 length and subunit number.

3. In the last two paragraphs of the section “Key interactions within the Erlin1/2 complex,” the authors use Figure 3e to illustrate the interaction of the Erlin CTDs with their surrounding hydrophobic environment. However, Figure 3e suffers from overlapping side chains and residue labels, which may confuse readers. I suggest the authors revise Figure 3e by splitting it into two panels, each showing the CTDs of Erlin1 and Erlin2 separately.

We thank the reviewer for the suggestion. We separated the information in Fig. 3e into two images: one emphasizing the locations of each CT (top), and the other showing hydrophobic packing, with the residues of distinct subunits colored differently (bottom).

Additionally, in the description, the authors refer to the interaction with “CC2 and β 9 of the heterodimer to the right, when viewed from the outside of the cage.” This phrasing may be unclear, as different readers could interpret “left/right” differently, even with the view angle specified as “the outside of the cage.” Using terms like “clockwise” or “counter-clockwise” might provide more clarity.

We adjusted these terms throughout the manuscript and added directional indications in relevant figure panels (Fig. 3d-f and 6g,h) to facilitate interpretation.

4. In the section “Positions of disease-linked Erlin1/2 mutations”, the authors only described the locations of the pathogenic mutations and provided some structural insight into the consequences of those mutations. Readers might benefit from a deep discussion on the biological impact of those mutations and how these impacts would cause disease.

We agree that understanding the molecular basis of how mutations in the Erlin proteins lead to disease would be ideal. However, in the absence of a well-defined function and reproducible functional assays for these complexes, as discussed above, including additional discussion regarding these mutants would be pure speculation. Importantly, based on the work conducted in our study, we can only speak rigorously regarding the location of these mutants in the context of the complex structure.

5. In Supplementary Figure 2, the reported particle counts and their corresponding percentages in the processing workflow appear inconsistent; please verify and correct these values.

We apologize for the confusion. The inconsistency came from two additional 3D classification steps after the 3D classification job shown in the original pipeline in Supplementary Fig. 2. We have added the results from the primary subsequent classification step to Supplementary Fig. 2 that further reduced particle numbers (to 41,900), while a following classification step only removed only ~10% of additional ‘junk’ particles (to 37,658). We also added the refinement result from these particles with C13 symmetry applied before they were subjected to C26 refinement and symmetry expansion.

Reviewer #2

Gao et al. present elegant cryo-EM structures of human organellar SPFH-family protein complexes, specifically Erlin1/2 (localized to the endoplasmic reticulum) and PHB1/2 (localized to the mitochondrial inner membrane). These SPFH proteins are key organizers of membrane microdomains and modulators of membrane protein function within their respective organelles.

Among the many important findings, the authors reveal that the Erlin1/2 complex assembles into a striking 26-subunit structure. Several disease-associated mutations in Erlin map to structurally critical regions of the complex, providing mechanistic insight into the pathogenesis of Erlin-related disorders. The authors also resolve the structure of the PHB1/2 complex, which forms a dimer of 22-subunit rings—comprised of 11 PHB1/PHB2 heterodimers—resulting in a dome-like assembly. While previous studies have linked the PHB complex to mitochondrial matrix AAA protease regulation, the current structures do not reveal associated protease densities, suggesting that these interactions may be transient, sub-stoichiometric, or spatially segregated. Nonetheless, the structures of the PHB1/2 complexes offer substantial new insight into their potential scaffolding and regulatory functions in mitochondrial organization.

Overall, this is a well-executed and impactful study that significantly advances our understanding of SPFH-family protein architecture and function. I recommend publication in Nature Communications without further revisions.

We thank the reviewer for their assessment.

Minor comments:

1. In the section describing the generation of the PHB1/2 complex for cryo-EM (Methods and Supplementary Fig. 6), the authors used an N-terminally Strep-tagged PHB1 construct stably expressed in Expi293F cells, followed by transient expression of untagged PHB2. I am curious whether the authors have performed localization studies—such as immunofluorescence—to confirm that the tagged PHB1 localizes correctly to mitochondria. It is important to rule out the possibility that the N-terminal tag interferes with mitochondrial import, potentially leading to mislocalization to the ER or plasma membrane. Even a brief comment on this point would be helpful to clarify this.

We performed immunofluorescence to confirm that N-terminally Strep-tagged PHB1 still co-localizes with a mitochondrial marker, and not with an ER marker (Reviewer Fig. 4).

Reviewer Fig. 4. Immunofluorescence of Strep-PHB1. Immunofluorescence of N-terminally Strep-tagged PHB1 (via an antibody against the Strep II tag; magenta) with the mitochondrial protein TOM20 (green, left panels) or the ER protein CANX (green, right panels) in cells without (bottom panels) or with lentiviral transduction (top panels) for stable Strep-PHB1 expression. Scale bar, 10 μ m. Note: specific staining of Strep-PHB1 in transduced cells, which colocalizes with TOM20 but not CANX.

2. Given the high-resolution structures, were any lipid densities observed or identified? While lipidomics is beyond the scope of this study, it could be highly informative in future work to determine whether specific lipids—such as cardiolipin or cholesterol—are associated with these complexes. Even speculative comments on potential lipid interactions based on observed densities would provide a valuable reference point for advancing our understanding of how these complexes interact with their membrane environments.

We thank the reviewer for this suggestion. We did observe several additional densities that proved challenging to conclusively assign. We now describe them in the text and in new Supplementary Fig. 3f and new Supplementary Fig. 9d. First, in the Erlin1/2 complex, we observed a density associated with every SPFH1 domain on the outside of the complex (new Supplementary Fig. 3f). Because the local resolution of this membrane-proximal region is lower, we did not feel that we could confidently model anything into this density. However, we note that a concurrent structural study (Yan et al. *bioRxiv* doi: 10.1101/2025.06.14.659634) saw a similar density at the same binding site and proposes that it may be a lipid headgroup, for example, of PI. The density in their map likely appears better resolved because they applied symmetry to their refinement, while we did not apply symmetry to our final refinement of a subset of symmetry-expanded particles. Second, in the PHB1/2 complex, we observe additional density between the SPFH2 domains of PHB1 and PHB2 that may involve (a possibly heterogeneous mixture of) modification(s) of Cys69 of PHB1 (new Supplementary Fig. 9d). However, mass spectrometry

analysis did not reveal a convincing singular modification on the peptide containing this reactive cysteine relative to a C69A PHB1 mutant. We also observe densities near the membrane–SPFH1 domain interface of the PHB1/2 complex (new Supplementary Fig. 9d), particularly one that may be coordinated by polar residues of PHB2.

Reviewer #3:

We thank the reviewer for their input.

Reply to Reviewers

All reviewers supported publication of the revised manuscript. None of their comments, repeated below, require additional changes or response.

Reviewer #1

I have carefully reviewed the authors' responses to my previous comments and suggestions, as well as the revisions made to the manuscript. I find that the authors have satisfactorily addressed the concerns raised in the original review, and the revised version is substantially improved in both clarity and coherence. I therefore support the publication of this work in Nature Communications.

Reviewer #2:

All of my comments have been addressed in the revision.

Reviewer #3:
